

# Coupled Ca and inorganic carbon uptake suggested by magnesium and sulfur incorporation in foraminiferal calcite

Inge van Dijk[1,2], Christine Barras[1], Lennart Jan de Nooijer[2], Aurélia Mouret[1], Esmee Geerken[2], Shai Oron[3], Gert-Jan Reichart[1,4]

[1]LPG UMR CNRS 6112, University of Angers, UFR Sciences, 2 bd Lavoisier 49045, Angers Cedex 01, France.
[2]NIOZ Royal Institute for Sea Research, Department of Ocean Systems (OCS), and Utrecht University, Postbus 59, 1790 AB Den Burg, The Netherlands.
[3]Charney School of Marine Sciences, University of Haifa, Israel.
[4]Utrecht University, Faculty of Geosciences, Budapestlaan 4, 3584 CD Utrecht, The Netherlands.

*Correspondence to*: Inge van Dijk (Inge.van.Dijk@nioz.nl)

**Abstract.** Shell chemistry of foraminiferal carbonate proves to be useful in reconstructing past ocean conditions. A new addition to the proxy toolbox is the ratio of sulfur (S) to calcium (Ca) in foraminiferal shells, reflecting the ratio of $SO_4^{2-}$ to $CO_3^{2-}$ in seawater. When comparing species, the amount of $SO_4^{2-}$ incorporated, and therefore the S/Ca of the shell, increases with increasing magnesium (Mg) content. The uptake of $SO_4^{2-}$ in foraminiferal calcite is likely coupled to carbon uptake, while

the incorporation of Mg is more likely related to Ca uptake since this element substitutes Ca in the crystal lattice. The relation between S and Mg incorporation in foraminiferal calcite therefore offers the opportunity to investigate the timing of processes involved in Ca and carbon uptake. To understand how foraminiferal S/Ca is related to Mg/Ca, we analyzed the concentration and within-shell distribution of S/Ca of three benthic species with different shell chemistry: *Ammonia tepida*, *Bulimina marginata* and *Amphistegina lessonii*. Furthermore, we investigated the link between Mg/Ca and S/Ca across species and the

potential influence of temperature on foraminiferal S/Ca. We observed that S/Ca is positively correlated with Mg/Ca on microscale within specimens, as well as between and within species. In contrast, when shell Mg/Ca increases with temperature, foraminiferal S/Ca values remain similar. We evaluate our findings in the light of previously proposed biomineralization models and abiological processes involved during calcite precipitation. Although all kinds of processes, including crystal lattice distortion and element speciation at the site of calcification, may contribute to changes in the amount of S and Mg that

is ultimately incorporated in foraminiferal calcite, these processes do not explain the consistent co-variation between Mg/Ca and S/Ca values. We observe that groups of foraminifera with different calcification pathways, e.g. hyaline versus porcelaneous species, show characteristic values for S/Ca and Mg/Ca, which might be linked to a different calcium and carbon uptake mechanism in porcelaneous and hyaline foraminifera. Whereas Mg incorporation is linked to the Ca-pump, S is linked to carbonate ion concentration via proton pumping. The fact that we observe coupled behavior of S and Mg, within specimens

and between species suggests that proton pumping and Ca pumping are intrinsically coupled across scales.



# 1. Introduction

The elemental and isotopic composition of foraminiferal calcium carbonate shells reflect seawater chemistry, and is therefore widely used to reconstruct specific marine environmental conditions. Besides the potential of Mg/Ca and $\delta^{18}O$ to reconstruct seawater temperature, currently available proxies permit reconstruction of part of the marine inorganic carbon system (Beerling

and Royer, 2011; Hönisch and Hemming, 2005). One of the most recent additions to the proxy tool box is the sulfur to calcium ratio (S/Ca) values of foraminiferal shells. In both abiogenic and biogenic carbonates, sulfur is mainly present in the form of $SO_4^{2-}$, where it substitutes $CO_3^{2-}$ (Pingitore et al., 1995; Perrin et al., 2017). S/Ca is correlated to the ratio of $SO_4^{2-}$ and $CO_3^{2-}$ in seawater in both inorganic carbonates (Fernández-Díaz et al., 2010) as well as in foraminiferal calcite (Paris et al., 2014; van Dijk et al., 2017a). However, the few calibrations on foraminifera currently available are for the species *Amphistegina*

*gibbosa* and *Sorites marginalis* and show species-specific offsets: the amount of $SO_4^{2-}$ incorporated, and therefore the S/Ca, increases with increasing Mg content (van Dijk et al., 2017a). Coupled concentrations of S and Mg across species could be due to 1) increased incorporation of S as a response to elevated crystal lattice strain due to higher concentrations of other elements, like Mg (Mucci and Morse, 1983; Evans et al., 2015), or 2) coupled transport of elements to the site of calcification (van Dijk et al., 2017b). To understand species-specific effects and constrain the application of these proxy-relationships, it is

necessary to focus on understanding element incorporation in foraminiferal calcite during biomineralization, i.e. in this case apparent coupling of Mg and S uptake and incorporation into foraminiferal calcite. While incorporation of sulfur is likely coupled to carbon uptake, due to the relation between calcite S/Ca and seawater $SO_4^{2-}/CO_3^{2-}$, foraminiferal Mg/Ca more likely reflects processes related to Ca uptake. Therefore, studying the relation between these elements might provide insight in the processes involved and timing of uptake of elements during biomineralization.

Isotopic and element composition of the foraminiferal shell depends on the chemistry of the surrounding seawater, the chemistry of the fluid in the site of calcification and element-specific partitioning, which in inorganic experiments is known to rely on precipitation rate (Mucci, 1987; Lorens, 1981). Foraminiferal species either create a smooth porcelaneous shell, relatively rich in magnesium (> 4%wt Mg) or perforate hyaline shells by supposedly contrasting calcification pathways (e.g. Hemleben et al., 1986; de Nooijer et al., 2009). In perforate hyaline species, studies have shown that the site of calcification is

separated from the surrounding seawater by a protective envelope. Although observations for *Ammonia* suggest this envelope might not be closed at the start of calcification (Nagai et al., 2018), the carbonate chemistry at the site of calcification is proposed to be controlled by the foraminifer and e.g. characterized by a high internal and low external pH (de Nooijer et al., 2009; Glas et al., 2012; Toyofuku et al., 2017), as well as the chemistry of the calcification fluid itself (Erez, 2003; Bentov and Erez, 2006; Evans et al., 2018). Controlling the physico-chemical conditions at the site of calcification is necessary to

overcome inhibition of calcite nucleation and growth by sulfate and magnesium ions (e.g.Zeebe and Sanyal, 2002; Fernández-Díaz et al., 2010; Reddy and Nancollas, 1976) Removal or immobilization of these ions is therefore part of several proposed biomineralization models (Bentov and Erez, 2006; Bentov et al., 2009), whereas inhibition can also be overcome by increasing



the saturation state at the site of calcification without a strong control on ions that inhibit calcification (Zeebe and Sanyal, 2002; de Nooijer et al., 2009; Toyofuku et al., 2017).

Here we investigate the co-variation between magnesium content and sulfur incorporation in foraminifera. At the microscale-level this is done by analyzing the distribution of S and Mg within the shell wall, and at the species-level by comparing S/Ca

and Mg/Ca of different species covering a wide taxonomic range, including different calcification pathways. Furthermore, we measured S/Ca values of shells of *Amphistegina lessonii* grown at different temperatures to investigate the potential effect of temperature on foraminiferal element incorporation. By changing seawater temperature, incorporation of Mg in many species of foraminifera increases (for an overview of different species: Toyofuku et al., 2011) due to the empirical positive relationship between temperature and Mg/Ca in calcite (e.g. Nürnberg et al., 1996). We investigate the sulfur and magnesium incorporation

in *A. lessonii* as a function of temperature, as to our knowledge a temperature-Mg calibration is currently lacking for this species. All observations combined shows how Mg and S are incorporated in foraminiferal calcite across scales, and thereby provide new insights on element uptake and transport during foraminiferal biomineralization.

## 2. Methods and materials

### 2.1 Collection of foraminifera

Coral debris was collected from Burgers' Zoo in Arnhem, the Netherlands, by scuba diving. The corals from the Indo-Pacific Ocean and coral debris is rich in a wide range of tropical foraminiferal species from this region (Ernst et al., 2011). The seawater in this aquarium is maintained at near natural conditions (salinity, temperature and carbonate chemistry). Collected sediment was transported to the Royal Netherlands Institute of Sea Research (NIOZ), and stored in aerated small aquaria at room temperature. From this stock, living specimens of *Amphistegina lessonii* were collected from the coral debris for the

culture experiment. Viable specimens of *A. lessonii*, recognized by color, attachment to coral debris and pseudopodial activity, were isolated and stored per 5-10 specimens in 70 ml Petri dishes with 0.2 μm filtered North Atlantic surface seawater (salinity = 35.2) with the addition of 1 ml/L trace metal K mix (Guillard and Ryther, 1962). Furthermore, from the foraminiferal stock, specimens of other foraminifera species with either hyaline (*Heterostegina depressa* - in addition to *Amphistegina lessonii*) or porcelaneous (*Sorites orbiculus*, *Spiroculina angulata*, *Spiroloculina communis*, *Quinqueloculina pseudoreticulata* and

*Quinqueloculina* sp.) shells were picked to study species-specific incorporation of sulfur and magnesium in foraminiferal calcite.

### 2.2 Set-up controlled temperature experiment

After asexual reproduction events of isolated specimens of *A. lessonii* (about 1/3 of the specimens reproduced), the four most numerous generations (~40-80 specimens per reproduction event) of ~2 chambered juveniles were incubated in duplicate with

0.2 μm filtered North Atlantic surface seawater in 70ml tissue bottles with hydrophobic caps at three different experimental conditions, resulting in ~15-25 incubated clones per condition. Tissue bottles containing foraminifera were placed in either





one of three climate incubators, set to 21, 26 and 29 °C. Using tissue bottles with hydrophobic caps minimizes the amount of evaporation compared to culturing Petri dishes. Nevertheless, half of the culture medium was replaced every three days, during which salinity was measured. Salinity never deviated more than 0.3 units from the stock value of 35.2. Temperature within each incubator was monitored every 11 minutes by a temperate logger (Traceable Logger Trac, Maxi Thermal). The average

temperature for the three different conditions was 21.2 ±0.7°C, 26.3 ±0.3°C and 29.5±0.2°C (+/- 1 SD). Shelves within each incubator were equipped with in-house designed and manufactured LED shelves (full spectrum) to provide uniform light (par) conditions in the incubators. The LED lights were controlled by a time controller and set to 300 par (high-light condition) for a 12 h/12 h day/night cycle.

## 2.3. Foraminiferal calcite chemistry

### 2.3.1. Cleaning methods

Groups of selected species of Burgers' Zoo stock and *Amphistegina lessonii* from the controlled temperature experiment were cleaned before analysis of their shell chemistry. Specimens were transferred to acid-cleaned 0.5 ml PCR-tubes (TreffLab). For cleaning, we followed an adapted version of the protocol by Barker et al. (2003), described in van Dijk et al. (2017b). In short, to each vial, 250 µl of fresh prepared 1% $H_2O_2$ (buffered with 0.5 M $NH_4OH$) was added to remove organic matter. The vials

were heated for 10 min in a water bath at 95 °C, and placed in an ultrasonic bath (80 kHz, 50% power, degas function) for 30 s after which the oxidizing reagent was removed with 3 rinses with double de-ionized water. These steps (organic removal procedure by oxidation) were repeated twice. Foraminiferal samples were subsequently rinsed five times with ultrapure water and dried in a laminar flow cabinet. Specimens from Burgers' Zoo were set apart for analysis by sector field ICP-MS, while the specimens from the controlled temperature experiment as well as a small number of *A. lessonii* from the Burgers' Zoo

stock were placed on stubs for analysis by laser ablation (Reichart et al., 2003).

### 2.3.2. LA-Q-ICP-MS

Element concentrations of individual chambers of A. lessonii from the controlled temperature culture experiment as well as the Burgers' Zoo stock were measured by laser ablation quadrupole-inductively coupled mass spectrometer (LA-Q-ICP-MS), similar as described in a number of previous studies (van Dijk et al., 2017b; Geerken et al., 2018). In short, using an ArF

Excimer laser (Existar) with deep UV 193 nm wavelength and <4 ns pulse duration (NWR193UC, New Wave Research) with a circular spot of 80 µm, a repetition rate of 6 Hz and an energy density of ~1 J/cm2, individual chambers of foraminiferal shells were targeted. The resulting aerosol was transported from the helium environment in a dual-volume cell of ~1 cm3 (New Wave, TV2), to a Q-ICP-MS (iCap, Thermo Scientific) on a helium flow (700 ml/min). Before entering the torch, 400 ml/min Argon was added using a 10 cm house-made smoothing device. Nitrogen was not added to the carrier gas to enable accurate

measurement of 55Mn. Other monitored masses include 7Li, 23Na, 24Mg, 25Mg, 43Ca, 44Ca, 88Sr, 137Ba, with a total cycle time of 140 ms. Calibration was performed against MACS-3, a pressed powder carbonate (synthetic calcium) standard, with



[43]Ca as an internal standard. SRM NIST 610 and 612 glass standards were measured in triplicate at the end of each series (energy density of 5±0.1 J/cm[2]). Accuracy and precision per element are reported in Table 1.

In total, 441 chambers were measured; 142 ablations on 59 specimens for a temperature of 21.2°C, 189 ablations on 63 specimens for 26.3°C and 110 on 42 specimens for T=29.5°C. Element concentrations were calculated by integrating individual laser-ablation profiles using an adapted version of the data reduction software SILLS (Signal Integration for Laboratory Laser Systems; Guillong et al., 2008) package for MATLAB (Geerken et al., 2018; van Dijk et al., 2017b). Profiles were selected to avoid contamination of the outer or inner part of the foraminifera (for examples of profile selection, see e.g. Dueñas-Bohórquez et al., 2011; Mewes et al., 2014; van Dijk et al., 2017c). Average E/Ca per temperature conditions were calculated after removal of outliers (based on 1.5*Interquartile range). We applied a t-test to assess if E/Ca is different between temperature conditions using a bilateral test.

### 2.3.3. SF-ICP-MS

Grouped foraminifera from the controlled temperature experiment and the species from Burgers' Zoo stock (*Amphistegina lessonii, Heterostegina depressa, Sorites orbiculus*, *Spiroculina angulata*, *Spiroloculina communis*, *Quinqueloculina pseudoreticulata* and *Quinqueloculina* sp.) were analyzed for the sulfur content in their shells. Foraminifera from the controlled temperature experiment received an additional cleaning step (following the same procedure as van Dijk et al., 2017a), since they were previously fixed on a laser ablation stub with tape. These specimens were transferred in groups of ~approximately ten individuals to 0.5 ml acid-cleaned TreffLab PCR-tubes and rinsed three times with 750 µl of de-ionized water, during which the vials were transferred to the ultrasonic bath for 1 minute (80 kHz, 50% power, degas function). Subsequently, the samples were again placed in the ultrasonic bath for another minute after addition of 750 µl of suprapur methanol (Aristar). The solvent was removed and the samples were rinsed three times with ultrapure water. Vials were dried in a laminar flow cabinet.

Samples from the temperature experiment and from the Burgers' Zoo were measured on different occasions, using a slightly different analytical approach. For the samples of the temperature experiment, 150 µl of ultrapure 0.05 M $HNO_3$ (PlasmaPURE) was added to each vial to dissolve all foraminiferal calcite. A five second pre-scan for [43]Ca was performed on the SF-ICP-MS to determine the [Ca] in the dissolved foraminiferal calcite solutions and accordingly to these results, samples were diluted to obtain a solution with 40 ppm Ca. After the cones were preconditioned during 2 hours with 40 ppm pure $CaCO_3$, final solutions were measured again for ~170 seconds per sample. Samples were injected into the ICPMS using a microFAST MC system of ESI with a loop of 250 mm and a flow rate of 50 ml/min. For the sample set of the temperature experiment, masses [23]Na, [24]Mg, [32]S, [34]S and [43]Ca were analyzed in medium resolution to separate $^{16}O^{16}O$ from the $^{32}S$ peaks, and $^{18}O^{16}O$ from the $^{34}S$ peaks.

The samples from Burgers' Zoo were dissolved in 0.5 ml 0.1 M $HNO_3$ and diluted to 100 ppm Ca accordingly to the results of the Ca pre-scan. Elemental composition of the foraminifera was measured for a wide range of elements, including [23]Na, [24]Mg, [32]S, [34]S and [43]Ca at medium resolution. In total 46 isotopes were measured during six min at low, four min at medium and one minute at high resolution with a 300 ml/min flowrate using a peristaltic pump. For both sets of measurement, samples



were measured against six ratio calibration standards with a similar matrix. A drift standard was measured after every 5[th] sample. In addition to the foraminiferal samples, we measured several standards, including NFHS-1 (NIOZ Foraminifera House Standard; for details see Mezger et al., 2016) and JCp-1 (coral, *Porites* sp.; Okai et al., 2002) to monitor drift and the quality of the analyses. We used a ratio calibration method (de Villiers et al., 2002) to calculate foraminiferal S/Ca (mmol/mol).

**2.3.4. Electron probe micro-analysis**

Specimens of various foraminiferal species (*Ammonia tepida*, *Bulimina marginata*, *Amphistegina lessonii*) from a recently published culture study (*A. tepida* and *B. marginata* from Barras et al., 2018; *A. lessonii* from unpublished experiment with a set-up similar to Barras et al., 2018) were prepared for electron probe micro-analysis (EPMA) to investigate the intra-shell incorporation of sulfur and magnesium. These foraminifera were cultured under hypoxia (30% oxygen saturation) and

previously studied to investigate the Mn incorporation in foraminiferal calcite (for details and culture methodology, see Barras et al., 2018). Specimens of each species were embedded under vacuum in resin (2020 Araldite® resin by Huntsman International LLC) using 2.5 cm epoxy plugs. Samples were polished using increasingly finer sanding paper. In the final polishing step, a diamond emulsion with grains of 0.04 μm was used, resulting in exposure of cross section of chamber walls. After applying a carbon-coating, the samples were placed in the microprobe sample holder. After selection of target areas,

several small high-resolution maps (130x97 pixels) were analyzed at 12.0 kV in beam scan mode for different elements (Ca, Mn, Mg, S and Na) with a dwell time of 350 ms. In total, we analyzed between 12 and 15 maps per species, an overview of the number of maps per species can be found in Table S1.

All EPMA data was further processed using MATLAB, following similar protocols used by van Dijk et al. (2017a) and Geerken et al. (2018). In summary, pores and resin were excluded from the final maps by excluding areas where Ca levels

were below a certain level (mostly around <10,000 counts). The resulting concentration (level) maps were converted to semi-quantitative E/Ca$_{EPMA}$ by applying a calibration based on mineral standards (diopside for Ca, tephroite for Mn, forsterite for Mg, coelestine for S and jadeite for Na). We choose to report these ratios as E/Ca$_{EPMA}$ to distinguish this data from quantitative data obtained by e.g. LA- and SF-ICP-MS. Mn, Mg, S and Na matrices were divided by the Ca matrix, to allow for a semi-quantification of the counts to concentrations to obtain E/Ca$_{EPMA}$ (mmol/mol) maps.

For all successful EPMA maps, i.e. high-quality maps without distortion or charging during measurement, several rectangular areas perpendicular to the chamber wall (Fig. 1) were selected, thereafter called transect maps. In total we created 23, 15 and 16 transect maps for *A. tepida*, *B. marginata* and *A. lessonii,* respectively. From these transect maps, the average Mg/Ca$_{EPMA}$ and S/Ca$_{EPMA}$ values per column over the transect are plotted, resulting in a spatial distribution profile (Fig. 1D). The average values per column are plotted, to investigate the co-variation of S/Ca and Mg/Ca in the foraminiferal chamber wall (Fig. 1C;

S/Ca$_{EPMA}$ versus Mg/Ca$_{EPMA}$)

Only one transect per EPMA map is considered for further analysis of peak and base values, to avoid overrepresentation of one chamber. This resulted in 11, 8 and 10 transect maps with distribution profiles for *A. tepida*, *B. marginata* and *A. lessonii* respectively. For the resulting distribution profiles, we calculated and compared the average values of the maximum (high-



concentration bands; E/Ca$_{MAX}$) and minimum (low-concentration areas; E/Ca$_{MIN}$) of the first peak which is related to the primary organic sheet. The offset between maximum and minimum value (Δ max-min; Fig. 2) is calculated as absolute concentration difference (E/Ca$_{MAX}$ - E/Ca$_{MIN}$ in mmol/mol) and expressed as a peak factor (E/Ca$_{MAX}$ / E/Ca$_{MIN}$). This allows for comparison of maximum (band) and minimum (non-band) values of E/Ca between the three species.

## 5   3. Results

### 3.1. S/Ca and Mg/Ca of *Amphistegina lessonii* from controlled temperature conditions

S/Ca of *A. lessonii* cultured at 21.2-29.5°C is on average 1.48 mmol/mol, with no trend with increasing temperature. Mg/Ca increases with temperature, and LA and SF-ICP-MS measurements are in good agreement. Based on the SF-ICP-MS data Mg/Ca is on average 19.7 mmol/mol for 21.2°C, 25.5 for 26.3°C and 34.8 mmol/mol for 29.5°C (Fig. 3) For LA-ICP-MS (Fig.

S1 and Table S2) average values of Mg/Ca are 20.5±5.7 mmol/mol for 21.2°C, 24.9±4.4 for 26.3°C and 35.4±8.1 mmol/mol for 29.5°C, which is 97.2, 102.4 and 98.4% compared to the SF-ICP-MS data. Thus, Mg/Ca increases by 1.8 mmol/mol per 1°C in our studied temperature range. Element to calcium ratio of other elements, Na/Ca, Sr/Ca and Mn/Ca are presented in the supplementary information (Fig. S1 and Table S2).

### 3.2. S/Ca of foraminiferal species from an Indo-Pacific aquarium

Average values of S/Ca and Mg/Ca per species collected from the Burgers' Zoo aquarium, including values of *A. lessonii* from the controlled temperature experiment, are presented in Fig. 4 and Table S3. For porcelaneous species Mg/Ca and S/Ca are on average 148.9±14.9 mmol/mol and 9.4±1.2 mmol/mol, respectively, while hyaline species cover broader ranges, with Mg/Ca from 36.8-153.3 mmol/mol and S/Ca between 2.2-8.4 mmol/mol.

### 3.3. Intra-shell distribution of sulfur and magnesium

All three species studied here, *A. tepida*, *B. marginata* and *A. lessonii*, show alternating bands with high and low concentrations of Mg and S, but the absolute values of these minimum and maximum values differ between species (Fig 5). For all three species we observe a positive correlation between Mg/Ca$_{EPMA}$ and S/Ca$_{EPMA}$ values of the distribution profiles (example in Fig. 1C). Average coefficients of determination ($R^2$) is 0.60±0.2 for *A. tepida* (23 profiles), 0.52±0.17 for *B. marginata* (15 profiles) and 0.85±0.1 for *A. lessonii* (16 profiles). For all transects maps, the average S/Ca$_{EPMA}$ and Mg/Ca$_{EPMA}$ is calculated

and average S/Ca$_{EPMA}$ and Mg/Ca$_{EPMA}$ are respectively 1.7±0.1 and 3.9±0.2 mmol/mol for *A. tepida*, 1.8±0.2 and 5.5±0.4 mmol/mol for *B. marginata*, and 2.4±0.3 and 23.9±2.3 mmol/mol for *A. lessonii*. The values for S/Ca$_{EPMA}$ versus Mg/Ca$_{EPMA}$ of each transect map are plotted in Fig. 6. For all three studied species, values of S/Ca$_{EPMA}$ and Mg/Ca$_{EPMA}$ of individual transects are significantly positively related for the three species assuming a linear regression. Based on the slopes of the regressions, S/Ca$_{EPMA}$ increases with ~38-43% for *A. tepida* and *B. marginata* respectively, and ~9% for *A. lessonii* relative to



Mg/Ca$_{EPMA}$, from now on referred to as 'S/Ca-Mg/Ca slope'. *Amphistegina lessonii* has the highest S incorporation, but the least sensitive S/Ca-Mg/Ca slope (Fig. 6).

### 3.4 Peak-base analysis

Based on eleven peak analyses, *A. tepida* has both the lowest peak- and base values (E/Ca$_{PEAK}$/ E/Ca$_{BASE}$) with 4.9/3.5 and

2.1/1.5 mmol/mol for Mg/Ca$_{EPMA}$ and S/Ca$_{EPMA}$ respectively. For *B. marginata* we analyzed eight transects, with average peak and base values of 7.0/4.1 and 2.5/1.5 for Mg/Ca$_{EPMA}$ and S/Ca$_{EPMA}$ respectively. Based on ten transects, *A. lessonii* has the highest values for mmol/mol both S/Ca$_{EPMA}$ and Mg/Ca$_{EPMA}$ for peak and base, 56.6/10.2 and 6.9/1.9 mmol/mol for Mg/Ca$_{EPMA}$ and S/Ca$_{EPMA}$ respectively. This data is summarized in Fig. 7 and Table 2.

Peak factors of Mg/Ca$_{EPMA}$ and S/Ca$_{EPMA}$ are very similar within the species *A. tepida* and *B. marginata*, respectively between

1.4x (Mg/Ca$_{EPMA}$) and 1.5x (S/Ca) for *A. tepida* and 1.7x (both Mg/Ca$_{EPMA}$ and S/Ca$_{EPMA}$) for *B. marginata*. For *A. lessonii* the peak factor is much higher for both S/Ca$_{EPMA}$ (3.6x) and Mg/Ca$_{EPMA}$ (2.8x), indicating more pronounced peaks in the latter species. The peak and base value of S/Ca$_{EPMA}$ are very similar for both low Mg species; average base values are 1.5 mmol/mol and peak values are 2.1-2.5 mmol/mol. *Amphistegina lessonii* has slightly higher base values for S/Ca$_{EPMA}$ (1.9 mmol/mol), but average peak value of S/Ca$_{EPMA}$ is higher (6.9 mmol/mol). When comparing the difference between peak values of Mg/Ca and S/Ca, the S/Ca peak is 43% and 12% of the Mg/Ca peak for both respective species *A. tepida* and *A. lessonii*, which might

be reflected in the steeper slope of S/Ca-Mg/Ca relation for *A. tepida* observed in the transects (Fig. 6).

## 4. Discussion

### 4.1. Foraminiferal S/Ca and Mg/Ca as a function of temperature

Shell Mg/Ca of specimens of *A. lessonii* grown under controlled temperatures increases with 1.8 mmol/mol per °C (Fig. 3).

To our knowledge, this is the first Mg/Ca-temperature calibration for *Amphistegina* spp. In inorganic carbonate precipitation studies, temperature is suggested to increase the thermodynamic Mg partitioning coefficient (Katz, 1973), which is also often used to explain the positive correlation between Mg and temperature in foraminiferal calcite. This abiogenic effect of temperature is, however, not a sufficient explanation, and is therefore thought to be enhanced or modified by a biological processes (Branson et al., 2013). Studies on the distribution of Mg in foraminiferal calcite show that temperature modulates

the Mg/Ca values of both the high-concentration bands and the low-concentration baselines (Spero et al., 2015; Fehrenbacher et al., 2017; Geerken et al., in review). Although we observe an increase in S with Mg for different species, S incorporation does not increase over the temperature range studied here (Fig. 3B). Within the species *A. lessonii* we found a S/Ca-Mg/Ca slope of 9% (Fig. 6), which would translate to an increase in shell S/Ca of i.e. 0.9 mmol/mol, when Mg/Ca values increase from 20 to 35 mmol/mol over the studied temperature range (Fig. 3 and S. Fig. 1). The absence of an effect of temperature on

S/Ca suggests that the process responsible for increasing Mg, does not affect SO$_4^{2-}$ substitution for CO$_3^{2-}$ in the crystal lattice.



This implies that although not controlled by temperature itself, S and Mg incorporation are still somehow coupled during biomineralization.

### 4.2. Mg and S distribution in the foraminiferal shell

For all three species studied here we observe a positive correlation between Mg/Ca and S/Ca within chamber walls (intra-specimen variability; for an example, see Fig. 1C), between transects maps for each species (inter-specimen variability; Fig. 6) and between species (inter-species variability; Fig. 4). Both bands of high S and high Mg seem to be located close to organic linings, as shown previously for *A. gibbosa* (by EPMA; van Dijk et al., 2017a) and *A. lobifera* (by electron probe WDS; Erez, 2003). As shown in Fig. 1D, the relative intensity of Mg/Ca and S/Ca for different peaks in the same transect are not always similar (e.g. rightmost peaks in Mg/Ca and S/Ca are equally high, whereas the S/Ca peak in the middle of the chamber wall is much lower than that of Mg/Ca), suggesting that S and Mg are incorporated simultaneously, but their concentrations must be partly decoupled as well. For Mg/Ca, peak and base values are higher for *A. lessonii* than for the low Mg species *A. tepida* and *B. marginata* (Fig. 7). The higher Mg/Ca in shells of *A. lessonii* compared to the two low Mg species (*A. tepida* and *B. marginata*) appears to be caused by an increase in both peak and base concentrations in *A. lessonii* specimens, already suggested by Geerken et al. (in review). In contrast, the base values (i.e. non-band area) of S/Ca seems to be very similar for all the three species (1.5±0.1 mmol/mol for both *A. tepida* and *B. marginata* and 1.9±0.2 mmol/mol for *A. lessonii)* suggesting that the increase in S/Ca for species with higher Mg incorporation (Fig. 4) might be due to higher S-peaks in the S/Ca banding. By comparing S/Ca and Mg/Ca values within transects we can test whether there is a correlation of S/Ca and Mg/Ca between specimens of the same species (inter-specimen variability). Individual transects were compared for each of the three species, showing a significant positive correlation between S/Ca$_{EPMA}$ and Mg/Ca$_{EPMA}$ (Fig. 6). Based on the slopes of the regression, S/Ca increases with ~37 and ~39% for *A. tepida* and *B. marginata* respectively, and ~9% for *A. lessonii* relative to Mg/Ca. The larger benthic foraminifer *A. lessonii* has the highest S incorporation, but the least sensitive S/Ca-Mg/Ca slope. In all species, higher or lower average Mg/Ca and S/Ca values are likely caused by respectively more or less intense banding between specimens (i.e. inter-species variability). The peak of bands of Mg and S of the small benthic species (*A. tepida* and *B. marginata*) are much closer, peak S/Ca are 43 and 36% of Mg/Ca respectively in these species, than for *A. lessonii* in which the S/Ca peak is 12% of the Mg/Ca peak. This results in a higher relative increase of S/Ca with Mg/Ca for the small benthic species, and thus a steeper S/Ca-Mg/Ca slope. Still, although the slopes differ between these groups, S/Ca and Mg/Ca are consistently positively correlated.

### 4.3 What controls Mg$^{2+}$ and SO$_4^{2-}$ uptake?

Correlation between S/Ca and Mg/Ca in foraminiferal calcite (Fig. 6) might reflect i) precipitation processes, occurring at the crystal-solution interface (e.g. effects of lattice strain and crystal growth rate) or in the solution occupying the site of calcification (e.g. speciation of elements in seawater and the effect of elevated pH), ii) biomineralization-related processes, like a coupling of ion transport to the site of calcification, or iii) a combination of both. The amount of variability and unknowns



in combination with the lack of knowledge on crucial processes involved in foraminiferal calcification make it challenging to assess which processes are ultimately responsible for the uptake and incorporation of both sulfate and magnesium into foraminiferal calcite. However, the observed lack of a temperature effect on S incorporation in contrast to the major impact of temperature on Mg/Ca may render some explanations more likely than others.

### 4.3.1. Calcite precipitation and physico-chemical conditions

The observed link between S/Ca and Mg/Ca might be explained by investigating parameters involved in inorganic precipitation studies. Chemical processes operating at the crystal-solution interface or in the fluid contained in the site of calcification might give insights in the observed correlation between sulfate and magnesium incorporation in foraminiferal calcite, as well as the temperature effect on Mg incorporation. Magnesium ions in the parent solution have been found to increase the co-precipitation

of other elements (Okumura and Kitano, 1986). However, this is observed for alkali metal ions, which are in interstitial positions or substitute for $Ca^{2+}$ in the crystal lattice, while $SO_4^{2-}$ is hypothesized to exchange for $CO_3^{2-}$ ions (Pingitore et al., 1995; Perrin et al., 2017; Berry, 1998). Besides promoting co-precipitation, incorporation of magnesium in carbonate is suggested to cause strain on the crystal lattice, leading to distortion, and an increase in the incorporation of other alkaline elements (Mucci and Morse, 1983). This theory has been used to explain incorporation of certain elements, like $Na^+$ and $Sr^{2+}$

in larger benthic foraminifera (Evans et al., 2015). Besides the lack of study on the incorporation of sulfate with increasing Mg content, based on our data, the correlation between Mg/Ca and S/Ca cannot be explained (solely) by crystal lattice distortion. The lack of response of S/Ca to (temperature-induced) changes in Mg/Ca (Fig. 3) together with the similar base values of S/Ca for all three investigated species, while base values for Mg/Ca vary between species (Fig. 7), show that S/Ca and Mg/Ca are not always correlated, which should be the case with this hypothesis.

The effect of temperature on Mg/Ca has been comprehensively studied in inorganic carbonate, by controlled precipitation experiments (for a summary see Mucci, 1987). The last decades, several explanations have been proposed to explain the relation between temperature and Mg incorporation in inorganic carbonates. Firstly, the partitioning of certain elements in inorganic experiments heavily relies on precipitation rate (e.g. Lorens, 1981). This was suggested for Mg incorporation (Chilingar, 1962), which may indicate that the increase of foraminiferal Mg/Ca in our study could be explained by a positive

effect of temperature on precipitation rate. However, this was disputed by several studies (e.g. Mucci and Morse, 1983; Mucci et al., 1985), showing precipitation rate does not change with temperature, but is depending on the Mg/Ca ratio of the parent solution.

It has been proposed that calcium- and magnesium transport to the site of calcification requires complete or partial dehydration of these ions, an energy-consuming process that is influenced by temperature (Mucci, 1986; Morse et al., 2007; Arvidson and

Mackenzie, 2000). Re-hydration of these ions at the site of calcification may furthermore determine isotopic fractionation during calcium carbonate precipitation (Mavromatis et al., 2013). Since dehydration of magnesium ions costs less energy at higher temperatures, a positive effect of temperature on foraminiferal Mg/Ca may indicate an increased inward Mg-transport, whereas an increased selective removal of Mg at higher temperatures would result in a lower Mg/Ca at higher temperatures.



Since the latter is not observed, the effect of the (de)hydration of Mg ions is only likely in biomineralization models where Mg is not selectively removed, like the Trans Membrane Transport (TMT) mixing model (Nehrke et al., 2013). Although this explains the lack of a clear temperature effect on S/Ca-values, it does not explain coupled S/Ca and Mg/Ca behavior.

Speciation of elements in seawater as a function of carbonate chemistry parameters (e.g. $[CO_3^{2-}]$ and/or pH) has been proposed

as an explanation for the incorporation of Zn and U in foraminiferal calcite (Djogić and Branica, 1991; Keul et al., 2013; van Dijk et al., 2017c). The effect of temperature and pH on the activity or bioavailability of different chemical species of Mg and S in seawater has not been studied so far, but can be modelled using the software package PHREEQC (Parkhurst and Appelo, 1999) using the in-software llnl database and standard seawater composition. This allows us to test two different conditions: A) variable temperature with stable $pCO_2$ (current atmosphere) and salinity (35) and B) constant temperature and salinity

(25°C and 35 respectively) and increasing pH and stable alkalinity of 2300μmol/kg seawater (Fig. 8). Using CO2SYS, other carbon parameters including dissolved inorganic carbon (DIC) was calculated with K1 and K2 from of Lueker et al. (2000).

At variable temperature, the model shows a small decrease in the activity of $SO_4^{2-}$, while for Mg species, only activity of $MgCO_3$ complexes increases, which might suggest foraminifera use $MgCO_3$ complexes when calcifying, since activity ratio of $Mg^{2+}$ and $Ca^{2+}$ remains stable over this range. When temperature changes from 20 to 30°C, the activity of $SO_4^{2-}$ decreases

by 8% (due to a small increase in the activity of $MgSO_4^-$), while the activity of $MgCO_3$ increases by 28%. This could, in part, explain the lack and presence of an effect of temperature on foraminiferal S/Ca and Mg/Ca respectively, when assuming foraminifera incorporate this species of magnesium during biomineralization. Abundance of sulfur in the form of sulfate will decrease slightly with increasing temperature leading to lower S/Ca of the shell. For *Amphistegina*, the 8% decrease in activity of $SO_4^{2-}$ might counter balance the expected 9% increase in S/Ca with temperature, based on the S/Ca-Mg/Ca slope we

observed (Fig. 6), leading to no observed change in foraminifera S/Ca with temperature (Fig. 3). This might, in part, control the ultimate E/Ca if these species are used to precipitate the foraminiferal shell. During chamber addition, pH changes externally to the foraminifera (pH<8; Glas et al., 2012; Toyofuku et al., 2017) and internally at the site of calcification or in seawater vacuoles (de Nooijer et al., 2009). Availability of $SO_4^{2-}$ remains similar when pH changes, while presence of $MgCO_3$ increases with pH, especially when pH>8 (Fig. 8b), which argues in favor for a major role of $MgCO_3$ complexes. Still, when

comparing the amplitude in Mg/Ca from 21.2 to 29.5 °C, incorporation almost doubles (Mg/Ca increases ~72%; Fig. 3 and Table S2), whereas the change in relative abundance of $MgCO_3$ complexes increases only by 28%. Hence, although we cannot exclude a role of $MgCO_3$ complexes it does not explain the full range observed.

### 4.3.2. Element transport during biomineralization

Based on the transmembrane transport mixing model (TMT; Nehrke et al., 2013; Mewes et al., 2015), $Mg^{2+}$ might be

accidentally transported to the site of calcification by Ca-channels or –pumps as well as by passive transport (e.g. leakage, initial seawater enclosed at the site of calcification or seawater endocytosis), while $SO_4^{2-}$ would not be transported by the Ca-channels or -pumps. Only prior to or at the first stages of chamber formation, when the membrane is perhaps not fully closed (Nagai et al., 2018) and the fluid in the site of calcification (SOC) resembles seawater, $SO_4^{2-}$ is incorporated due to the relatively





high $[SO_4^{2-}]$ in seawater. The clear single peak at the start of the lamella as shown by the EPMA analysis, might indicate there is either much more sulfate present at the start of calcification, or the $CO_3^{2-}$ concentration is still low, and hence the $SO_4^{2-}$ to $CO_3^{2-}$ ratio.

In the seawater vacuolization (SWV) model (Bentov et al., 2009), the main source of ions is from the endocytosis of seawater.

The Mg/Ca of the fluid in these seawater vacuoles is lowered (to ~2 mol/mol; Evans et al., 2018), but it is not known if the sulfate concentration is regulated in these vacuoles, making it impossible to assess whether $Mg^{2+}$ and $SO_4^{2-}$ concentrations in the vacuoles are correlated. The similar and spatial stable baseline values for S/Ca for the three species studied here suggest that uptake or transport of $SO_4^{2-}$ to the site of calcification during the shell thickening phase of calcification is matched by incorporation of $SO_4^{2-}$ in the calcite, leading to a similar S/Ca of the non-band areas.

Considering both models individually, it is impossible to explain the correlation between $SO_4^{2-}$ and $Mg^{2+}$ incorporation. However, by considering certain constrains on element transport offered by both models, we can hypothesize which E/Ca might be characteristic for both end-member models to understand the relation of Mg/Ca and S/Ca on species-scale (Fig. 4). We consider three different processes, each resulting in a different E/Ca signature that foraminifera might employ to take up calcium and carbon:

    i)      SWV dominated: During endocytosis, $[Mg^{2+}]$ in the vacuoles will be actively lowered, while $[SO_4^{2-}]$ is hypothesized to remain at its original concentration.

    ii)     Ca channel-dominated: Due to the transport of $Ca^{2+}$ through Ca-channels to the site of calcification, Mg/Ca will be lowered, while there is no effect on the $SO_4/CO_3$ from which calcite is precipitated.

iii)    Proton pumping: Pumping of protons out of the SOC (Toyofuku et al., 2017) will increase its pH and shift the speciation of inorganic carbon towards $CO_3^{2-}$. This increase in $[CO_3^{2-}]$ will lead to lower $SO_4^{2-}/CO_3^{2-}$ values in the site of calcification.

All currently available data from previous culture and field studies in which values for both S/Ca and Mg/Ca data are available

(van Dijk et al., 2017a; Mezger et al., in prep.) combined with values from the controlled temperature study as well as the Burgers' Zoo specimens as well as the semi-quantitative data from EPMA analysis (paragraph 3.3), are presented in Fig. 9 and Table S4. Because carbonate associated sulfur in foraminifera is incorporated as $SO_4^{2-}$ and the ratio between Ca and $CO_3$ is 1, S/Ca can be converted (1:1) and expressed as $SO_4^{2-}/CO_3^{2-}$.

In general, S incorporation increases linearly with increasing Mg content, with a S/Ca-Mg/Ca slope of ~6%, i.e. if Mg/Ca

increases by 1 mmol/mol, S/Ca increases by 0.06 mmol/mol. Interestingly, no offset is observed between porcelaneous and high-Mg hyaline species, as has been noted before for other elements (as observed for e.g. Na/Ca; van Dijk et al., 2017b). Both groups seem to have a characteristic chemical signature: hyaline species have in general low S and Mg incorporation (except for the high Mg hyaline species, like *Heterostegina depressa*), while porcelaneous species have high S and Mg values. The combination of low S/Ca and Mg/Ca for planktonic and small benthic hyaline foraminifera might be the result of the



combination of Ca transport and proton pumping, two processes already linked in low-Mg foraminifer *Ammonia tepida* (Toyofuku et al., 2017). High-Mg hyaline and porcelaneous foraminifera seem to occupy the SWV dominated region, which is supported by both the observation of seawater vacuoles in hyaline species and the calcification pathway of porcelaneous species, which is suggested to take place intracellular in the form of calcite needle formation in vacuole-like structures. This

data highlights the fundamentally different calcification pathways proposed before for these two groups (e.g. Berthold, 1976; Hemleben et al., 1986). However, due to the limited amount of constrains on both models it is currently difficult to fully assess the impact of both calcification pathways on element incorporation, and therefore we cannot exclude other factors which might be responsible for the correlation between Mg and S incorporation.

### 4.3.4. Sulfate at the site of calcification

Species specific differences in the relative contribution of SWV and TMT might provide an explanation of our results. However, while this could give insights in the incorporation of S and Mg as a function of temperature and explain species-specific differences, we did not consider the inhibition effect of sulfate and the probable non-classical calcification pathway foraminifera utilize to create their shell (Jacob et al., 2017). Sulfate is a known inhibitor for precipitation of calcite (e.g. Manoli and Dalas, 2000; Kitano, 1962), but does play a role in the transformation from amorphous calcium carbonate into vaterite

(see Bots et al., 2012 and references therein) and stabilizes this metastable carbonate phase vaterite when solution $SO_4^{2-}:CO_3^{2-}$ > 1 (Fernández-Díaz et al., 2010). A recent study has proposed that certain species of planktonic foraminifera create their shell by a pathway involving vaterite phases that transform ultimately to calcite (Jacob et al., 2017), which might suggest the $SO_4^{2-}:CO_3^{2-}$ at the site of calcification is > 1 when precipitation of the carbonate shell commences. Just prior to formation of a new chamber, the sulfate concentration at the SOC is probably similar to that in seawater, assuming the calcification fluid is

composed of either a small volume of seawater enclosed by the protective envelop separating the SOC from seawater or by seawater vacuoles (SWV model).

Laboratory experiments have revealed that the internal pH of a foraminifera is elevated to ≥9 at the start of shell formation (de Nooijer et al., 2009) due to proton pumping (Toyofuku et al., 2017), which lowers the pH in the microenvironment surrounding the foraminifer (Glas et al., 2012). When assuming the $SO_4^{2-}$ and inorganic carbonate concentration at SOC is equal to natural

seawater at 400 ppm $CO_2$ (~2650 mg/L [$SO_4^{2-}$], ~2100 µmol/L DIC), the elevation of internal pH to 9 creates a $SO_4^{2-}:CO_3^{2-}$ of ~25, leading to the stabilization of vaterite and a band enriched in $SO_4^{2-}$ close to the primary organic sheet, which we observe in the chamber wall distribution of all three of our species (Fig. 5). The $SO_4^{2-}:CO_3^{2-}$ likely decreases during the thickening of the chamber wall, due to continuous active pumping of protons out of the site of calcification (Toyofuku et al., 2017) (Fig. 7). However, to confirm these hypotheses, a more precise characterization of the calcification fluid's chemistry is necessary.



## 5. Conclusions

Systematics in the incorporation of different elements in foraminiferal shells can be used to test calcification models and hence processes involved in precipitation of calcium carbonates. Our dataset, including both hyaline and porcelaneous species of foraminifera with a wide range of shell Mg content, shows a positive relation between Mg and S incorporated in their shells. This correlation can be found on species-scale, but also between specimens of the same species and on microscale in the heterogeneous distribution in the shell wall. In contrast, we find no effect of temperature on the S/Ca values of foraminiferal calcite, even though shell Mg/Ca increases. The lack of an observed temperature effect on S/Ca for *Amphistegina lessonii* might be due to the decrease in activity of sulfate with temperature, counterbalancing the increase in S/Ca due to increasing shell Mg/Ca. Nevertheless, the lack of certain crucial key factors, like the chemistry at the site of calcification, make it difficult to understand fully the pathway of both elements during calcification. The differences observed for the three species highlights the diversity of and variation in processes involved in biomineralization in foraminifera. Also mechanisms suggested for inorganic precipitation, like e.g. crystal lattice strain, precipitation rate and ion dehydration fail to independently explain our findings in full. Likewise, it is at this moment challenging to reconcile all these observations using a single calcification model. Comparing our data with existing biomineralization models implies that, irrespective of the model, foraminiferal Mg/Ca and S/Ca are governed by two different, but coupled, pathways. Mg/Ca is primarily affected by Ca-transport and passive transport, while S/Ca is mainly governed by proton pump intensity and passive transport. The observed patterns imply that these pathways are spatially and temporally linked, and hence that all hyaline and porcelaneous foraminifera, even though fundamentally differing in shell characteristics, actively take up calcium and carbon in a coupled process.

**Acknowledgements.** This research was carried out under the program of Darwin Centre for Biogeosciences (project 3020 by GJR) and Netherlands Earth System Science Center (NESSC; grant no. 024.002.001 by GJR). Further financial support comes from University Bretagne Loire and Angers Loire Metropole (project MOXY by CMB) and the French national program EC2CO-LEFE (project MANGA 2D by AM) and CNRS (INSU project MANGA 2D by AM). Great thanks to Wim Boer for support with LA-Q-ICP-MS and SF-ICP-MS measurements and both Tilly Bouten and Sergei Matveev are thanked for technical support with the EPMA analysis.

 

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





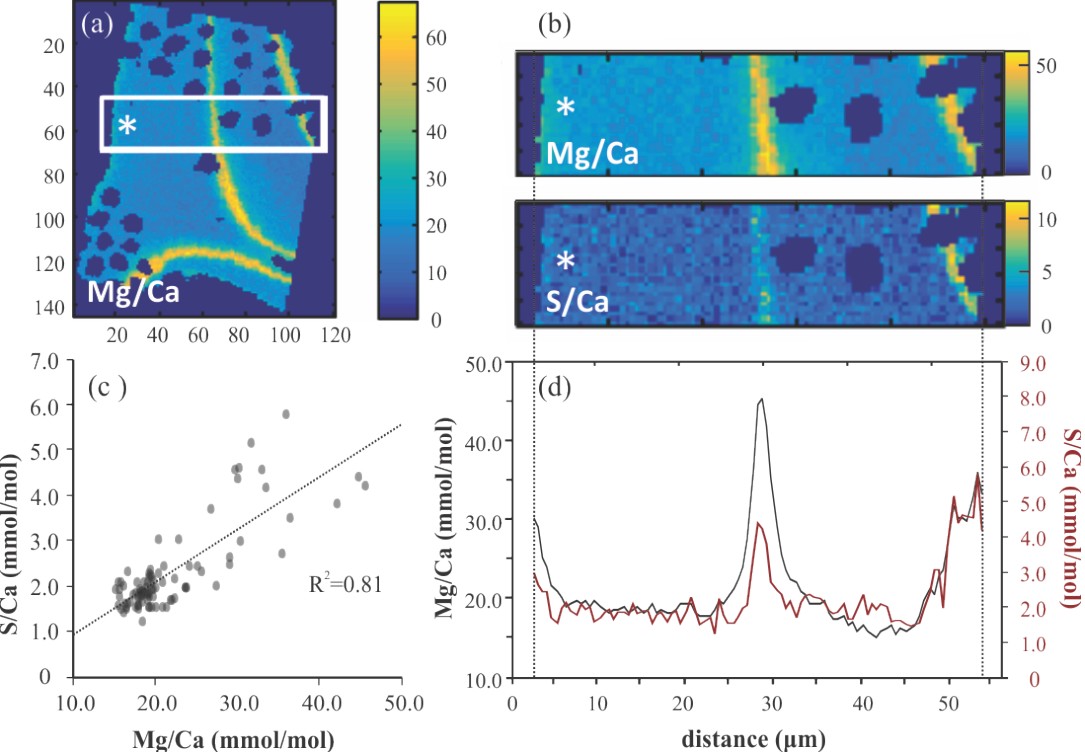

**Figure 1: Example of EPMA data treatment: A) General overview of Mg/Ca$_{EPMA}$ map with transect selection, B) Selection of the map from A) to isolate a transect of Mg/Ca$_{EPMA}$ (top) and S/Ca$_{EPMA}$ (bottom), C) Average S/Ca$_{EPMA}$ and Mg/Ca$_{EPMA}$ values plotted per column with the R$^2$ for the regression analysis, D) Average E/Ca$_{EPMA}$ of transect, Mg/Ca$_{EPMA}$ in black, S/Ca$_{EPMA}$ in red. Asterisks indicate outer side of the foraminiferal shell. Color scale is 'parula', in-software.**

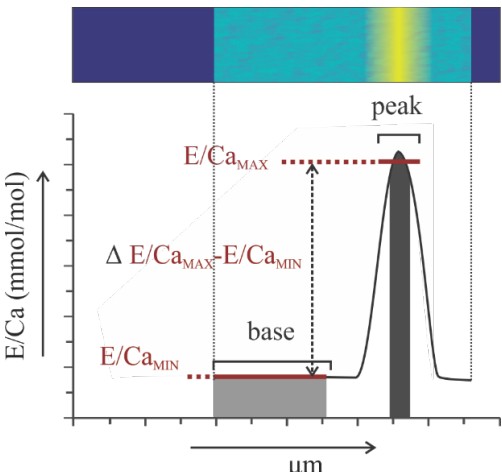

**Figure 2: Illustration showing theoretical transect map and resulting distribution profile with area of selection for peak-base analysis.**



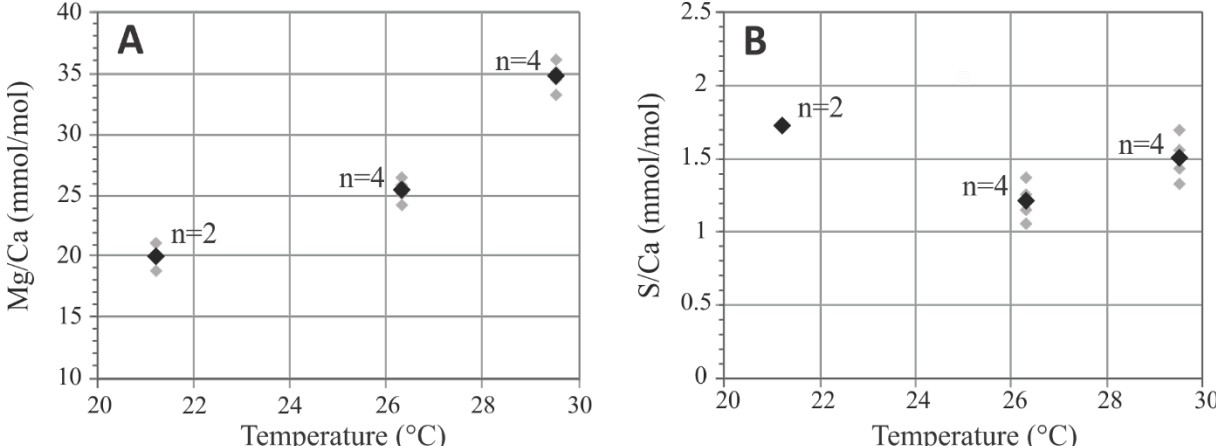

**Figure 3. Mg/Ca (A) and S/Ca (B) of *A. lessonii* from controlled temperature experiment. Sector field ICP-MS measurements (grey symbols) and average values (black symbols) of grouped specimens. n is number of measurements.**

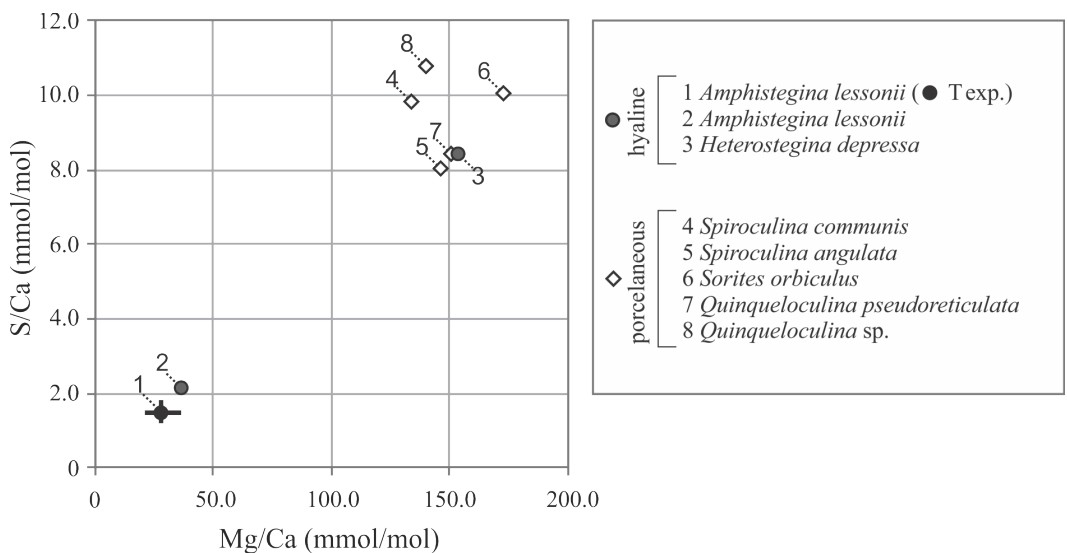

**Figure 4: Compilation of S/Ca versus Mg/Ca of foraminifera from the aquarium in Burgers' Zoo (circles: hyaline species; diamonds: porcelaneous species). Average value of the foraminifera from the temperature-controlled experiment are given in black, with maximum and minimum ranges. Numbers indicate species.**

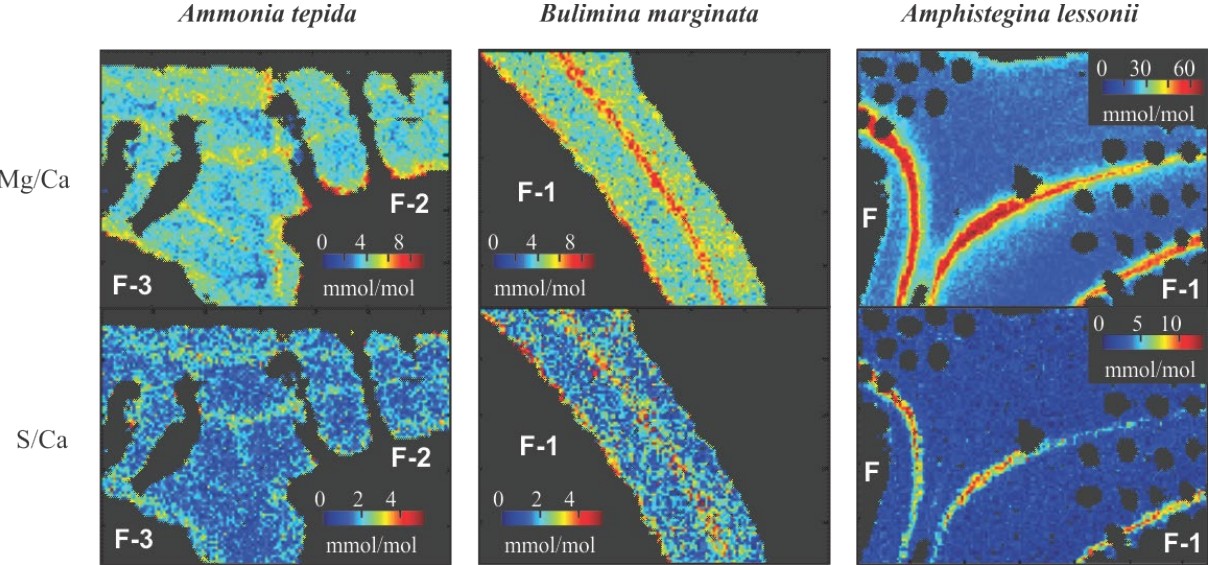

**Figure 5: Typical distribution of Mg/Ca (top panels) and S/Ca (lower panels) in three species of foraminifera, *Ammonia tepida* (left panels), *Bulimina marginata* (middle panels) and *Amphistegina lessonii* (right panels). Chamber numbers are indicated in white, on the interior part of the shell.**

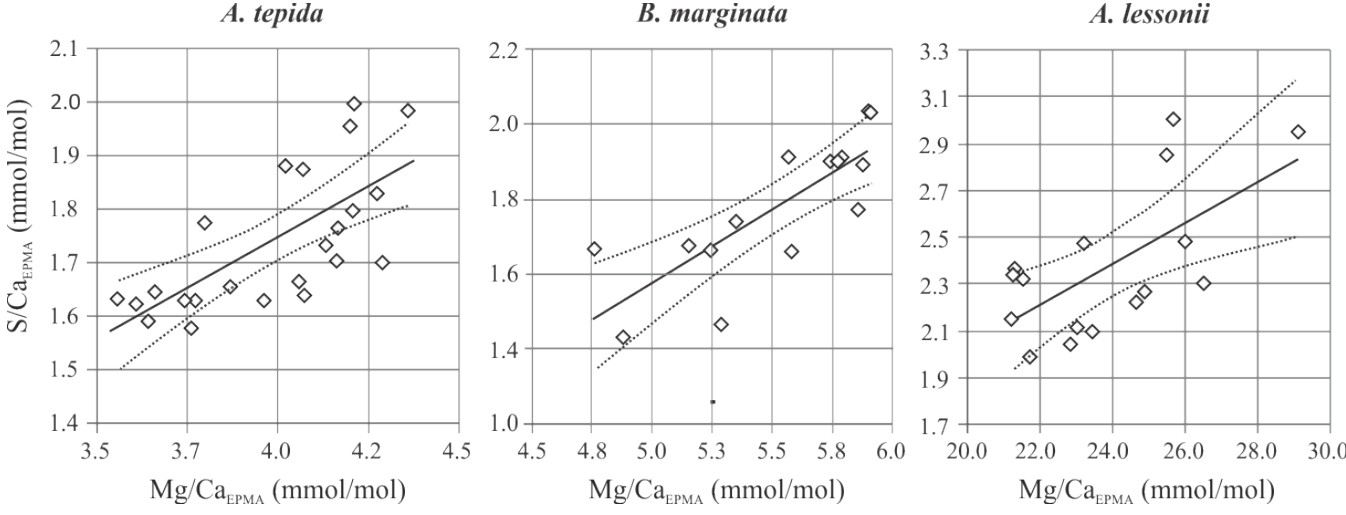

**Figure 6: S/Ca$_{EPMA}$ versus Mg/Ca$_{EPMA}$ values of the EPMA transects maps of different species. Every symbol represents the average S/Ca$_{EPMA}$ and Mg/Ca$_{EPMA}$ of a single transect map which are positively correlated for *Ammonia tepida* (n=24; S/Ca$_{EPMA}$=0.38*Mg/Ca$_{EPMA}$ +0.28 with R$^2$=0.47; p<0.0005), *Bulimina marginata* (n=15; S/Ca$_{EPMA}$=0.43*Mg/Ca$_{EPMA}$-0.18 with R$^2$=0.82; p<0.0005) and *Amphistegina lessonii* (n=16; S/Ca$_{EPMA}$=0.089*Mg/Ca$_{EPMA}$+0.26 with R$^2$=0.42; p<0.0025). NB. Confidence interval (95%) is indicated with dotted lines. Data is semi-quantitative and therefore expressed as E/Ca$_{EPMA}$, since calibration is performed against matrix unmatched mineral standards.**



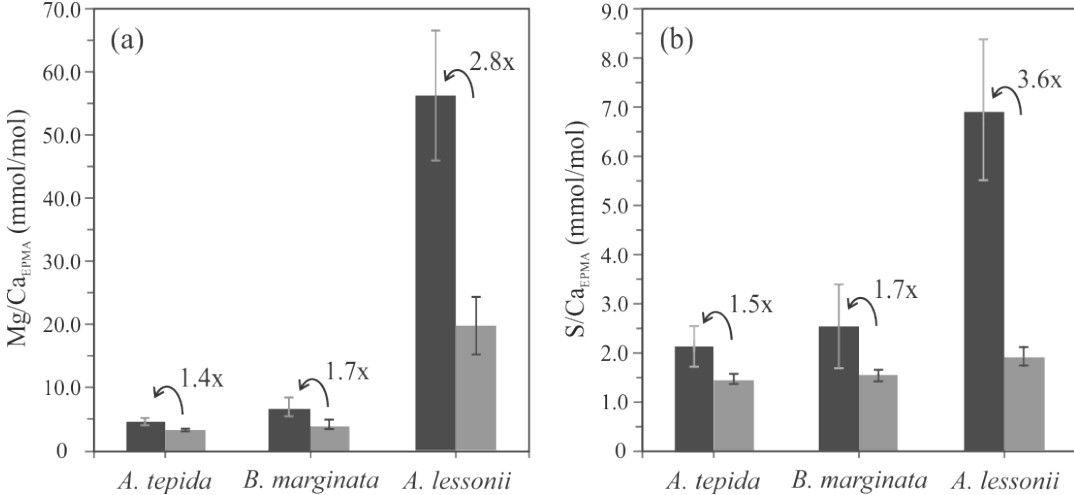

**Figure 7: Average peak (black bars) and base (grey bars) values (Mg/Ca$_{EPMA}$, panel A; S/Ca$_{EPMA}$, panel B) with 'peak factor' (E/Ca$_{MAX}$ / E/Ca$_{MIN}$; see methodology section 2.3.4 and Fig. 2) of three investigated species, low Mg species *A. tepida* and *B. marginata* and intermediate Mg species *A. lessonii*. Error bars indicate 2SD. For details, see Table 2.**

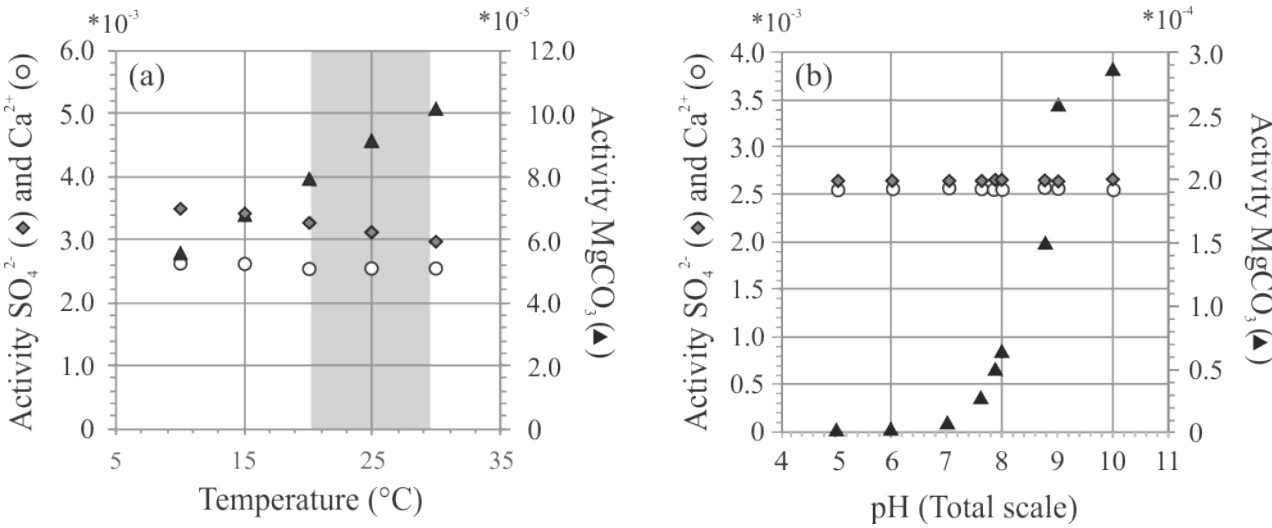

**Figure 8: Two exercises using PHREEQC showing activity of SO$_4^{2-}$, Ca$^{2+}$ and MgCO$_3$ at a) different temperatures (temperature range of controlled culture experiment indicated in grey) and b) different pH. Chosen pH range reflects external and internal pH shift during chamber formation (Glas et al., 2012; de Nooijer et al., 2009; Toyofuku et al., 2017).**





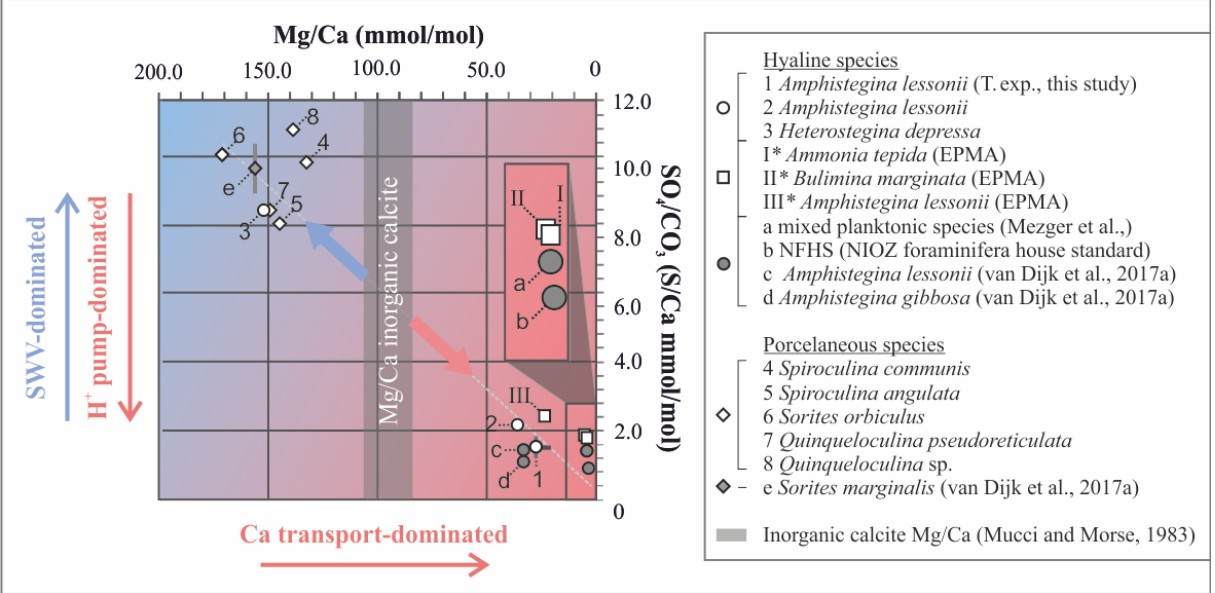

**Figure 9: Average values of S/Ca, expressed as $SO_4^{2-}/CO_3^{2-}$, versus Mg/Ca of different hyaline (circles) and porcelaneous (diamonds) species of this study (open symbols) and other studies (grey symbols), including values obtained for the NFHS (for a description see Mezger et al., 2016). Culture experiments with varying salinity (c), temperature (1) and $p$CO₂ (d,e), min and maximum ranges are indicated by grey bars, but in most cases fall within the symbol. Values of foraminiferal E/Ca can be found in Table S4. Values for Mg/Ca of inorganic precipitated calcite of Mucci and Morse, 1983 are indicated by the gray bar, SD is included. Linear regression based on all foraminifera-derived data points is S/Ca=0.06\*Mg/Ca+0.12, with $R^2$=0.94. Average values for transect maps (square symbols; for details see paragraph 3.3) are semi-quantitative and not included in the regression. Note the inversed axis compared to Fig. 4.**





| Element | MACS-3 (n=105) | NIST 610 (n=21) | | NIST 612 (n=20) | |
|---|---|---|---|---|---|
| | Precision (RSD, %) | Precision (RSD, %) | Accuracy (%) | Precision (RSD, %) | Accuracy (%) |
| $^{7}$Li | 5 | 1 | n/a | 4 | n/a |
| $^{23}$Na | 5 | 6 | 106.9 | 6 | 108.0 |
| $^{24}$Mg | 3 | 1 | 106.4 | 5 | 90.6 |
| $^{25}$Mg | 3 | 2 | 106.0 | 5 | 90.2 |
| $^{55}$Mn | 3 | 1 | 101.0 | 2 | 101.7 |
| $^{88}$Sr | 3 | 2 | 100.0 | 2 | 102.5 |
| $^{137}$Ba | 3 | 3 | n/a | 2 | n/a |

**Table 1: Accuracy (RSD of n measurements) and precision (% of reference value) of three standards, including the calibration standard MACS-3.**

| Species | Mg/Ca$_{EPMA}$ (mmol/mol) | | | | S/Ca$_{EPMA}$ (mmol/mol) | | | |
|---|---|---|---|---|---|---|---|---|
| | Peak | Base | Δ E/Ca$_{MAX}$ - E/Ca$_{MIN}$ | | Peak | Base | Δ E/Ca$_{MAX}$ - E/Ca$_{MIN}$ | |
| | | | mmol/mol | x | | | mmol/mol | x |
| *A. tepida* (n=11) | 4.9± 0.7 | 3.5± 0.2 | 1.5 | 1.4 | 2.1± 0.4 | 1.5± 0.1 | 0.7 | 1.5 |
| *B. marginata* (n=8) | 7.0± 1.4 | 4.1± 0.9 | 2.9 | 1.7 | 2.5± 0.9 | 1.5± 0.1 | 1.0 | 1.7 |
| *A. lessonii* (n=10) | 56.6± 10.2 | 20.2± 4.3 | 36.4 | 2.8 | 6.9± 1.4 | 1.9± 0.2 | 5.0 | 3.6 |

**Table 2: Results from peak-base analysis of *A. tepida* (n=11), *B. marginata* (n=8) and *A. lessonii* (n=10), with average peak (E/Ca$_{MAX}$)**
5 **and base (E/Ca$_{MIN}$) values in mmol/mol and ΔMin-Max parameters (mmol/mol; E/Ca$_{MAX}$ - E/Ca$_{MIN}$) and peak factor (x; E/Ca$_{MAX}$ / E/Ca$_{MIN}$); for details, see methodology section 2.3.4).**





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
