# Peer review of "Coupled Ca and inorganic carbon uptake suggested by magnesium and sulfur incorporation in foraminiferal calcite"

_Biogeosciences, 2018_

## Referee Comment (RC1) · Anonymous Referee #2 · 4 Jan 2019

Dear authors,

In your submitted article entitled "Coupled Ca and inorganic carbon uptake suggested by magnesium and sulfur incorporation in foraminiferal calcite" (van Dijke and collaborators), you present new in-situ and bulk data for Mg/Ca and S/Ca in cultured benthic foraminifera. Thanks to these new data you confirm the existence of a small scale correlation between S and Mg enrichment within individual tests, already known for some forams and now extended to three additional benthic species. You also demonstrate that this correlation seems to exist at the bulk scale. Eventually, you use bulk foram Mg/Ca vs. S/Ca data to infer new constraints on calcification mechanisms in different

porcelaneous and hyaline species. I find the paper of interest and I think that some rewriting could significantly improve it. TI thus recommend publication after moderate, yet necessary, modifications are performed.

Please find below my review of the paper.

_______________________________________________________________________________-

The submitted text provides interesting and intriguing new data. The methods are appropriate, the results require more descriptions. Data are not quite properly presented and errors not properly discussed. Individual error bars are not shown in the figures nor discussed in the text. Yet, for example, there is an error both and the X and Y axes in figure 3. In the results section, only the error on the averages is provided. What is the instrumental reproducibility? The external reproducibility? Improvements can be made regarding the writing (I provide a few suggestions in the detailed comments).

The path leading to the conclusion that Mg and S correlate at the bulk scale (EPMA profiles) could be improved. Obviously, even if there is a Mg-S correlation, the fact that there is a Mg-T and no S-T correlation is intriguing. Based on the data provided, the correlation could be described as significant for B. marginata but not for the other species analyzed here. Furthermore, when the authors present a correlation between EPMA-based Mg/Ca and S/Ca data, the values given in the text (page 7) do not match the values given in the caption of figure 6. Some questions could be addressed more explicitly. 1) do the EPMA correlation represent each one individual or more? 2) the individual errors should be reported on the figure, and the authors need to provide the MSWD. Are the correlations still "significant" when individual errors are taken into account? 3) the authors should define what they consider as "significant". Is there enough points for the p-value to be meaningful? 4) The forams used for those maps were grown under hypoxia. Can it alter the comparison? Why not use forams grown in regular oxic conditions similarly to those used in this manuscript? It seems to me that it would have made more sense to use for instance individuals from the temperature

experiments. What would the EPMA correlation look like if we took specimens from Fig. 3? That could help unravel the Mg-S, Mg-T correlation and lack of S-T correlations. I feel that the maps don't quite fit in the current story, and I wonder if that could be due to the fact that they were performed on specimens from other experiments.

Here follow some questions and thoughts about the interpretations the authors make of their data. -Why show only $MgCO_3$ and not Mg only activity in figure 8? Based on section 4.3.1, the $MgCO_3$ theory does not seem strongly supported by anything else than the current PHREEQC calculations. In the absence of reference further supporting the idea cited in the text, it should be much more developed). It would be more relevant to show all species as simple ions AND $CaCO_3$ and $MgCO_3$ to allow the reader to make their own opinion. I know that the atuhors can't plot every single dissolved species, but what they show seems too partial. -L. 33, the authors state that "an increased removal of Mg at higher temperatures would results in a lower Mg/Ca at higher temperature". That sounds logical as such, but they omit to explain why removal of Mg should increase at higher temperature in the first place. -Similarly, some possible mechanisms of correlation of Mg and S are not discussed. Tanaka et al., 2018 provide a thorough review of some mechanisms not considered here that are not to affect Mg in abiogenic calcite, such as the role played by organic matter, growth rate, saturation state rayleigh effect, or even modification of the calcifying fluid composition. The authors splits the processes as purely inorganic or biomineralization process-related (or a mixture of both) and do not include, for instance, the effects of organic matter on calcification. S is (too) often used as a mere way to track organic matter in the shell, without further discussion, but here this option is not even discussed. In section 4.2 the authors mention that S and Mg follow organic linings but then they do not follow up on that question. Yet, the presence of some type of OM can affect Mg (effect on $SO_4$ unknown) content (Mavromatis et al., 2017) and could provide a source of correlation. -The authors could mention the $SO_4/Ca$ Kd evaluated by Kitano et al., (1975) or Busenberg and Plummer (1985) in their figure 9 or in the discussion. -P. 12,L.5, L. 16 : why assume that sulfate is constant? That seems unlikely as we know that calcite

cannot precipitate at seawater sulfate concentrations. (This point is partially adressed in section 4.3.4, so it would probably make more sense to move up that section). In addition, the vacuole pH is modified (Bentov et al., 2009), so the SO4/CO3 ratio would necessarily be modified. Therefore this process needs to be reassessed. The Mg/Ca ratio might be lowered (not "Mg" as stated line 16), but the SO4/CO3 ratio will certainly be modified as well regardless of what happens to sulfate. -P. 12 L.17: I also disagree with this vector. Is it possible to Ca++ to cross a membrane and have no proton exchange? -P. 13 L.15-16: my understanding of Fernandez-Diaz (2010)'s paper is different. In this study, calcite+vaterite precipitates no matter what the SO4/CO3 ratio is. However, vaterite needs not be stable (see Jacobs et al. 2017: vaterite evolves to calcite by dissolution-reprecipitation). Fernandez-diaz and collaborators state that vaterite "transform into the more stable calcite via dissolution-reprecipitation although such transformation is hindered when the SO4/CO3 ratio in the fluid is higher than 1.3". This would put a higher limit on the sulfate content of the fluid in which the dissolution-reprecipitation needs to happen, maybe not necessarily a lower limit as suggested by the authors of the current manuscript.

As a result, it seems that figure 9 needs more constraints in order to make a solid case. The questions that are not solved by that figure are: -models often attempt to explain E/Ca ratios and therefore don't consider the fate or role of protons (eg. Langer et al., 2006; Nehrke et al., 2013) and sulfate is almost never considered. Combining them to explain Mg/Ca vs. SO4/CO3 would therefore more thinking about the role played by protons, a common way to ensure electroneutrality across a membrane when pumping or channeling ions or what happens to sulfate therefore seems possibly problematic. -does it work also for the EPMA profile-scale correlations? This scale does not seem to be interpreted in the text -does it make sense to calculate a R2 in figure 9? In the end, it is almost like a linear correlation between two (group of) points... -if the authors invoke vaterite as the precipitating polymoprh to solve the problem of sulfate inhibition on calcite precipitation, why use inorganic calcite in this figure?
* * *
Detailed comments P. 1 L. 14 substitutes for (and later in the text) L. 16 we analyzed the bulk concentration (overall in the text it would be good to ensure lack of ambiguity between bulk and in-situ analyses) L. 24 consistent small-scale covariation (same comment as above)

P.2 L.7 substitutes for L. 31 no "." before Removal

P.3 L.3 do the authors really investigate the co-varaition between Mg content and S incorporation? It seems to me that they investigate the co-variation between Mg and S contents to provide new constraint on biocalcification. The sentence should be improved.

P.4 L.5 temperature errors should be reported on fig. 3 Section 2.3: check exponents in isotopes. The name of the instruments are lacking. The laser is provided, not the various MS or EP. what are the blanks? the instrumental backgrounds? The instrumental error? The reproducibility of the data? The whole section needs more solid writing on the data quality and statistics. Section 2.3.3 signification of SF-ICP-MS not provided (could be done line 10)

P. 7 L. 23-28: the R values do not match that of figure 7 and the errors provided are not homogeneous. The p-values are just mentioned in the caption but no explanation is provided in the text. The notion of significance should be defined.

P.9 L.7 it seems worth referring here to Paris et al. (2014) nano-SIMS data as well, which do show that with much higher precision, the bands are not quite identical, at least in O. universa.

P. 10 L. 17 is the temperature-induced increase of Mg due to crystal lattice distortion change? It has too, in order to use this as an argument here, but it does not seem to be the case based on the explanations provided in the text.

P. 11 L.26 The conclusion sentence should be rewritten. It seems strange to write down

that it cannot be excluded, when the authors themselves are the one suggesting it.

Figure 9 caption: L.2 species of foraminifera?

---

## Referee Comment (RC2) · Anonymous Referee #1 · 7 Jan 2019

The manuscript Coupled Ca and inorganic carbon uptake suggested by magnesium and sulfur incorporation in foraminiferal calcite by van Dijk et al. presents both spatially-resolved (EPMA, LA-ICPMS) and bulk (SF-ICPMS) Mg/Ca and S/Ca data from several different benthic foraminifera, including hyaline and porcelaneous species. The aim is to examine the controls on shell Mg/Ca and S/Ca ratios in the context of different biomineralisation models, and to assess which process(es) might result in the observed intra-shell heterogeneity in the incorporation of these elements. The data, especially the EPMA maps are interesting and warrant publication, particularly because S/Ca has potential as a carbonate system proxy, yet relatively few S/Ca data exist for foraminifera. My main concern is that it is difficult to interpret the results as the relevant seawater

carbonate system data were not reported, so I recommend including those data and reframing the relevant parts of the discussion in terms of the relationship between the shell S/Ca and seawater $SO_4^{2-}$/$CO_3^{2-}$ ratios.

Major comments

- As far as possible, please report carbonate system details for both the culture experiment and the aquarium. What I found puzzling is that van Dijk et al. [2017] showed that S/Ca principally depends on the seawater [$CO_3^{2-}$], an exciting result, yet this is mostly not discussed in this manuscript. For example, in order to make sense of the data in Fig. 3B, we really need to know the [$CO_3^{2-}$] of the different cultures. If they do not conform to a (temperature-driven?) S/Ca-[$CO_3^{2-}$] relationship, then why, and how does this impact the conclusions of van Dijk et al. [2017]? Likewise, on page 12, line 24 onwards (Fig. 9), the shell $SO_4^{2-}$/$CO_3^{2-}$ ratio is discussed, but what we are interested in is the shell $SO_4^{2-}$/$CO_3^{2-}$ ratio as a function of the seawater $SO_4^{2-}$/$CO_3^{2-}$ ratio. It would be more informative to replot the data in this way.

- Throughout the manuscript, I was confused about what the term 'coupled' means. Is the implication that the Mg/Ca and $SO_4^{2-}$/$CO_3^{2-}$ ratios covary in the EPMA maps because these ratios in the calcifying space are (coincidentally?) being coregulated? If so, I recommend explaining more clearly what the inference is regarding how these ratios are modified through the process of chamber formation. In other places, the covariation ('coupling') of Mg and $SO_4^{2-}$ when comparing different species is mentioned, and so it is sometimes ambiguous whether covariation within a chamber, or within foraminiferal calcite in general is being discussed. Is the argument that similar processes are operating on all scales (between species/within a chamber etc.)?

- I appreciate that overall the authors have made an effort to consider how these data could fit into both the vacuolisation and TMT model. I would nonetheless urge more careful phrasing in places. For example, line 27 on page 1 states 'Mg incorporation is linked to the Ca-pump', but there is no strong evidence here for the TMT model

here, and the previously published evidence has been disputed. Fig. 9 should also be reconsidered. Low-Mg foraminifera are not all 'Ca transport-dominated', indeed it is very difficult to see how the concentration of most trace elements could be reconciled with the TMT model. Perhaps Ammonia does differ in this respect from the planktonics, but a low shell Mg/Ca ratio does not necessarily imply Ca TMT. Also rephrase line 18, page 2, line 15, page 14 (Mg-transport is also possible/likely), line 18, page 14 (the phrase 'actively take up' is ambiguous but certainly not all foraminifera transport Ca which is what I think most would understand from this statement).

Minor comments

- Page 1, line 24. Rephrase or remove the word 'consistent'. If the behaviour of Mg and SO42- incorporation is strictly consistent then surely some of the inorganic processes mentioned earlier in the sentence could explain what you observe.

- Somewhere in the introduction it would be useful to state how big the SO42 ion is compared to CO32-. Would lattice distortion from Mg incorporation be expected to favour SO42- incorporation?

- Page 2, line 21. Consider reversing the sentence; the chemistry of the shell depends in part on the chemistry of the fluid at the calcification site, which in turn likely depends on the ambient seawater.

- Page 2, line 31. What does 'immobilization of these ions' mean?

- Page 3, line 1. Please clarify 'without a strong control on ions that inhibit calcification'. Do you mean that increasing DIC does not have a large effect on the speciation of most ions?

- Page 4, line 6. I must be missing something obvious, but what does '(par)' mean?

- Page 4, line 31. Why was MACS-3 was used as the calibration standard? I understand the benefits of matrix-matching, but it has been argued that carbonate standardisation using NIST produces accurate results, and the issue with MACS-3 is that it is

not as homogeneous as NIST610 [see e.g. Jochum et al., 2012], which is also borne out by the data in Tab. 1.

- Page 6, line 1. Please be specific instead of simply stating 'similar'.

- Section 2.3.3. Please state accuracy and precision data for the SF S/Ca analysis, and how these were determined.

- Page 6, line 8. Again, please be specific. Was the set-up similar but different to that of Barras et al. [2018]? If so, in what way?

- Page 7, lines 11-12. I understand that it's difficult to assess the most appropriate regression form from three data points, but it would be interesting to report the exponential slope here too given that it appears the slope may be exponential from the available data.

- Section 3.4, Tab. 2, and Fig. 6. Clarify exactly what these data represent. Is each EPMA map from a different specimen, or different chambers of the same specimen? Does each data point in Fig. 6 represent the average Mg/Ca and S/Ca ratios of all data within each transect map? Similarly, on page 9, lines 23-27, how repeatable are these values? Do the percentages represent the average of several maps from several specimens? This would be much easier to follow if there was a supplementary figure showing the location of all maps/transects used to calculate each data point in Fig. 6, or add a column to Tab. S1 stating how many specimens and which chambers the transects are from.

- Page 8, line 11. On line 7 the peak and base ratios are quoted as 56.5 and 10.2, which would equate to a ratio of 5.5, not 2.8.

- Page 8, lines 12-14. This is repetition of the first paragraph in this section.

- Page 8, line 20. Although the data are somewhat challenging to interpret, there are technically Mg/Ca-temperature data for Amphistegina reported by Raja et al. [2007] doi: 10.1029/2006GC001478.

- Page 8, line 28. How is 0.9 derived? I calculate (35-20)*0.09 = 1.4 mmol/mol.

- Page 9, lines 1-2. How does this follow from the previous sentence? From Fig. 3 it appears that Mg and $SO_4^{2-}$ are not necessarily coupled, why do these data imply that they are?

- Page 9, line 10. I don't think 'incorporated simultaneously' is the right terminology. Rather, the higher concentration bands are located in a similar place.

- Page 10, lines 9-15. As it is written, it reads as if the mechanism for increased Mg/Ca resulting in increased alkali metal incorporation differs from that of the alkali earths. My understanding is that this is not necessarily the case, rather lattice distortion can result in increased incorporation into both lattice and interstitial sites (depending on ionic radius).

- Page 10, lines 23-27. I don't doubt that precipitation rate may be more sensitive to seawater Mg/Ca than temperature, but surely temperature will affect rate to some degree, if only because of the effect of temperature on carbon speciation through the temperature dependence of $K_W$.

- Page 10, line 32. I suggest removing this sentence. There is no observational or theoretical evidence for inward-directed Mg transport, and it is difficult to see what the purpose of this would be.

- Page 11, lines 1-3. The (de)hydration of any ion during attachment is a passive process depending on e.g. growth rate and the chemistry of the calcifying space. Why is it only likely in one of the biomineralization models?

- Page 11, lines 12-15 and line 24. Given that $MgCO_3$ is a small proportion of total Mg it seems unlikely that this is the explanation.

- Page 11, lines 17-20. I don't follow the logic here. It reads as if the argument here is that the relationship between sulphate and temperature is counteracted either by the increased shell S/Ca being driven by the increased shell Mg/Ca, or that sulphur is

actively transported to the calcification site to a greater degree at higher temperature. However, the first of these explanations is discounted elsewhere (e.g. page 14, line 12) and I do not see the mechanistic basis for the other based on the data presented here. To phrase it another way, surely the slope in Fig. 6 is being driven by the width or number of the co-located high-Mg, high-S bands in each transect, and unless the proportion of these changes as a function of temperature (does it?), why would the slope in Fig. 6 counteract a temperature-driven change in the activity of sulphate?

- Page 12, line 5. Evans et al. [2018] calculate that the Mg/Ca$_{sw}$ ratio at the calcification site of low-Mg foraminifera is <0.1 mol/mol, not 2 mol/mol as stated.

- Page 12, line 27. Perhaps a bit picky, but it is better approximated to Ca/(CO$_3^{2-}$ + SO$_4^{2-}$) = 1.

- Page 13, lines 16-18. I think this could be phrased more strongly, it is hard to see that the SO$_4^{2-}$/CO$_3^{2-}$ ratio is less than one given ∼30 mM [SO$_4^{2-}$], unless the DIC concentration is very high in the calcifying space.

- Page 13, line 22. It may well be species-specific, but note that the calcification site pH is not necessarily greater than 9 [∼8.75 according to Bentov et al., 2009].

- Page 24, lines 24-28. In both biomineralization models, pH is elevated in the calcifying space (or vacuole) in order to promote carbon concentration, and presumably the two processes occur simultaneously (what would be the benefit of separating them?). We don't know precisely to what extent this takes place or at what time, so I understand the reason for calculating it in this way, however my recommendation would be to rephrase this sentence as a constraint on the maximum calcification site SO$_4^{2-}$/CO$_3^{2-}$ ratio, given that the assumption of seawater DIC is probably not correct.

- Page 14, line 13. I don't think anyone has suggested that hyaline and porcelaneous foraminifera are characterised by the same biomineralization model.

- Fig. 1. State which species/treatment this map is from.

- Fig. 3. Please clarify whether the grey symbols represent repeat measurement of the same solution or different groups of 10 foraminifera.

- Fig 9. Proton pumping is a feature of both biomineralization models, so why should the arrow for 'SWV-dominated' be in the opposite direction to 'H+ pump-dominated'?

- Tab. 1 could be moved to the supplement.

Typos

- Page 2, line 31. There is a full stop missing after the parenthesis.

- Page 5, lines 28 and 33. Presumably it should read $\mu$l/min.

- Page 6, line 13. An emulsion refers to two immiscible liquids, replace with suspension.

- Page 7, line 23. Replace 'is' with 'are'.

---

## Author Response (AR1)

*Anonymous reviewer #1,*

*Thanks for the thorough reading of our manuscript and your constructive comments. Below, all our comments are listed in italics. We try to answer all specific comments raised by your review.*

Major comments

- As far as possible, please report carbonate system details for both the culture experiment and the aquarium. What I found puzzling is that van Dijk et al. [2017] showed that S/Ca principally depends on the seawater [CO32-], an exciting result, yet this is mostly not discussed in this manuscript. For example, in order to make sense of the data in Fig. 3B, we really need to know the [CO32-] of the different cultures. If they do not conform to a (temperature-driven?) S/Ca-[CO32-] relationship, then why, and how does this impact the conclusions of van Dijk et al. [2017]?

*For calculating the full carbonate system and thus CO32- we need to determine 2 parameters. As the aim of the study was not relate S and CO32- we (unfortunately) did not monitor e.g. alkalinity and DIC. We did occasionally measure pH of the culture media inside the culture bottle. However, beside the low number of measurements, pH is difficult to measure accurately. Still, we measured $pCO_2$ within the incubator, which was around 800 ppm. For the natural seawater we used, we assume a TA of ~2300 umol/L (extrapolated from salinity). Using the CO2sys software, we estimate the [CO32-] of the culture media to be between 115 and 145 umol/L over this temperature range. The experiment was specifically designed to study the effect of temperature on element incorporation. Hence, we focus the discussion on inter- and intra-specimen variation in S/Ca and Mg/Ca.*

*We used CO2SYS to calculate the expected change in CO32- due to differences in the used temperatures (assuming a constant alkalinity of 2300 umol/L and pCO2 of 800ppm). The difference in temperatures between treatments (21 and 29 degrees) accounts for a change in $CO_3^{2-}$ of ~30 umol/L. This offset between treatments is relatively small compared to the sensitivity of foraminiferal S/Ca to $[CO_3^{2-}]$. and would result in a change in S/Ca of 6% or ~0.08 mmol/mol (van Dijk et al., 2017, EPSL). Hence the S/Ca-[CO32-] relationship is not sensitive enough to detect such variability between treatments (Fig. 3). Therefore, changes due to differences in [CO32-] are most likely below the detection of this proxy.*

*Even though our experiment was not designed to determine the impact of carbonate ion on S incorporation, and taking into account the limited control we have on the sea water carbonate chemistry of the experiments, values plot in the same order of magnitude as those of van Dijk et al., 2017 calibration, see the figure (R1: In red, average values for* Amphistegina gibbosa *(van Dijk et al., 2017, EPSL). In yellow, average values for* Amphistegina lessonii *from this study) for your reference.*

Likewise, on page 12, line 24 onwards (Fig. 9), the shell SO42-/CO32- ratio is discussed, but what we are interested in is the shell SO42-/CO32- ratio as a function of the seawater SO42-/CO32- ratio. It would be more informative to replot the data in this way.

*Although intriguing, we were not able to plot the data this way, since SO42-/CO32- is not known for all conditions. However, the reviewer provides a good point, we can use shell S/Ca to unravel SO42-/CO32- of the calcifying fluid at the site of calcification, which will be added to the last paragraph of the discussion.*

*To be added to MS: "This implies that the S/Ca distribution in the foraminiferal chamber wall may reflect a change in $SO_4^{2-}/CO_3^{2-}$ of the calcifying fluid in the site of calcification (SOC) during precipitation of the shell wall. Assuming a stable D during calcification (E.g. Dx1000= 0.013; Busenberg and Plummer, 1985), $SO_4^{2-}/CO_3^{2-}$ at the SOC would be a factor of 3.6 higher during the*

*thin, high-concentration band (with an S/Ca of 6.9 mmol/mol; Fig. 7) compared to the broader, low-concentration band (with an S/Ca of 1.9 mmol/mol). This decrease by a factor of 3.6 could be due to an increase in $[CO_3^{2-}]$ and/or a decrease in $[SO_4^{2-}]$ during precipitation. The latter could be the result of inclusion of small amounts of sulfate in the SOC at the beginning of chamber formation and ongoing incorporation of sulfate in the foraminiferal calcite. However, since the S/Ca is not decreasing towards the outer side of the shell in the low-concentration band, the former process, i.e. increasing $CO_3^{2-}$ , might be more likely. An increase of $CO_3^{2-}$ at the SOC from the first stage of chamber formation (high in S/Ca) to the broader second part (low in S/Ca) could be caused by an increase in internal pH due to proton pumping (Toyofuku et al., 2017). The band of high S/Ca would then be precipitated when proton pumping has not yet reached its maximum rate and the internal pH is still rising (Glas et al., 2013). However, to confirm this hypothesis, a more precise characterization of the calcification fluid's chemistry is necessary."*

- Throughout the manuscript, I was confused about what the term 'coupled' means. Is the implication that the Mg/Ca and SO42-/CO32- ratios covary in the EPMA maps because these ratios in the calcifying space are (coincidentally?) being coregulated? If so, I recommend explaining more clearly what the inference is regarding how these ratios are modified through the process of chamber formation. In other places, the covariation ('coupling') of Mg and SO42- when comparing different species is mentioned, and so it is sometimes ambiguous whether covariation within a chamber, or within foraminiferal calcite in general is being discussed. Is the argument that similar processes are operating on all scales (between species/within a chamber etc.)?

*We tried to clear up the nomenclature used in the manuscript. We replaced coupled with coregulation and covariation.*

- I appreciate that overall the authors have made an effort to consider how these data could fit into both the vacuolisation and TMT model. I would nonetheless urge more careful phrasing in places. For example, line 27 on page 1 states 'Mg incorporation is linked to the Ca-pump', but there is no strong evidence here for the TMT model here, and the previously published evidence has been disputed. Fig. 9 should also be reconsidered. Low-Mg foraminifera are not all 'Ca transport-dominated', indeed it is very difficult to see how the concentration of most trace elements could be reconciled with the TMT model. Perhaps Ammonia does differ in this respect from the planktonics, but a low shell Mg/Ca ratio does not necessarily imply Ca TMT. Also rephrase line 18, page 2, line 15, page 14 (Mg-transport is also possible/likely), line 18, page 14 (the phrase 'actively take up' is ambiguous but certainly not all foraminifera transport Ca which is what I think most would understand from this statement).

*We rephrased these sentences, to also include differences in Mg transport out of the SOC as a process to change shell Mg/Ca. We removed only the word 'actively' from Page 14, line 18, since also with vacuolization, carbon and calcium are taken up together. In figure 9 we slightly modified the vector of SWV-dominated (see also comments reviewer 2).*

Minor comments

- Page 1, line 24. Rephrase or remove the word 'consistent'. If the behaviour of Mg and SO42-incorporation is strictly consistent then surely some of the inorganic processes mentioned earlier in the sentence could explain what you observe.

*This sentence is now rephrased.*

- Somewhere in the introduction it would be useful to state how big the SO42 ion is compared to CO32-. Would lattice distortion from Mg incorporation be expected to favour SO42- incorporation?

*This is now information added to the introduction based on the radii published in Jenkins&Thahur (1979). They state that SO42- ions are larger than CO32- ions, which would mean Mg distortion might increase the incorporation of SO42-.*

- Page 2, line 21. Consider reversing the sentence; the chemistry of the shell depends in part on the chemistry of the fluid at the calcification site, which in turn likely depends on the ambient seawater.

*This sentence is now rephrased.*

- Page 2, line 31. What does 'immobilization of these ions' mean?

*Elements might not only be physically removed, but also unavailable in terms of e.g. speciation/complexation. We changed 'immobilization' for 'unavailability'.*

- Page 3, line 1. Please clarify 'without a strong control on ions that inhibit calcification'. Do you mean that increasing DIC does not have a large effect on the speciation of most ions?

*We rephrased: '..without needing removal of ions that inhibit calcification'*

Page 4, line 6. I must be missing something obvious, but what does '(par)' mean?

*PAR = Photosynthetically active radiation ($\mu mol$ of photons $m^2\ s^{-2}$), which equals to high light conditions. We stated this now more clearly in the revised manuscript.*

- Page 4, line 31. Why was MACS-3 was used as the calibration standard? I understand the benefits of matrix-matching, but it has been argued that carbonate standardisation using NIST produces accurate results, and the issue with MACS-3 is that it is not as homogeneous as NIST610 [see e.g. Jochum et al., 2012], which is also borne out by the data in Tab. 1.

*We agree MACS-3 is less homogeneous than NIST610, but the concentrations of various elements, including Ca, Mg, Na, are more close to the concentrations found in foraminiferal calcite. Furthermore, NIST standards are extremely rich in sodium, and therefore ablation leads to a memory effect, increasing the Na background for the next ~30 measurements, changing the detection limit of this element considerably. Therefore, the multiple measurement of NIST at the start of the session decreases the quality of Na data.*

*All in all, the MACS-3 element composition approaches the foraminiferal values, which leads to a more robust calibration (less extrapolation) of the sample values. Since the precision for MACS3 is still 5% or lower, we believe this standard is the better option/trade off and gives a more realistic indication of potential inaccuracies. We added this shortly in section 2.3.2.*

*In manuscript: "We choose MACS-3 as a calibration standard, since element composition approaches the foraminiferal values closer than that of NIST 610 or 612 and therefore aids a more robust calibration, even though the MACS-3 is slightly less homogeneous (see precisions listed in Table 1)."*

- Page 6, line 1. Please be specific instead of simply stating 'similar'.

*We changed this to 'matching' to avoid confusion.*

- Section 2.3.3. Please state accuracy and precision data for the SF S/Ca analysis, and how these were determined.

*This are now added to the revised version of the manuscript (see also comments reviewer 2) in section 2.3.3.*

- Page 6, line 8. Again, please be specific. Was the set-up similar but different to that of Barras et al. [2018]? If so, in what way?

*We used an identical set-up, the only difference was the temperature (25°C) and light cycle (12hr/12hr). The similarity and differences between experiments is now more clearly stated in the text (section 2.3.4)*

*In manuscript: "Specimens of various foraminiferal species (Ammonia tepida, Bulimina marginata) from a recently published culture study (Barras et al., 2018), and Amphistegina lessonii cultured in the same culture set-up, were prepared for electron probe micro-analysis (EPMA) to investigate the intra-shell incorporation of sulfur and magnesium. These foraminifera were cultured under hypoxia (30% oxygen saturation) in controlled stable conditions and previously studied to investigate the Mn incorporation in foraminiferal calcite (for details and culture methodology, see Barras et al., 2018). Ammonia tepida and Bulimina marginata were cultured at 12°C, while specimens of Amphistegina lessonii were grown at 25°C. For the latter species, the set-up was equipped with a light system with 12 hr/12 hr light cycle."*

- Page 7, lines 11-12. I understand that it's difficult to assess the most appropriate regression form from three data points, but it would be interesting to report the exponential slope here too given that it appears the slope may be exponential from the available data.

*We included values for a linear regression as well as an exponential regression to the revised text to accommodate the reviewer.*

- Section 3.4, Tab. 2, and Fig. 6. Clarify exactly what these data represent. Is each EPMA map from a different specimen, or different chambers of the same specimen? Does each data point in Fig. 6 represent the average Mg/Ca and S/Ca ratios of all data within each transect map? Similarly, on page 9, lines 23-27, how repeatable are these values? Do the percentages represent the average of several maps from several specimens? This would be much easier to follow if there was a supplementary figure showing the location of all maps/transects used to calculate each data point in Fig. 6, or add a column to Tab. S1 stating how many specimens and which chambers the transects are from.

*We agree with the reviewer that we can be more clear about the locations of the EPMA maps and transects. We therefore added three figures and expanded a table in the supplementary information, showing the SEM overview pictures with EPMA targets of all three species. We furthermore added more details in section 2.3.4. and 3.3 about the number of transects per chamber/specimens/species, and which transects are presented in Fig 6 and 7.*

| Species | EPMA maps total n | n specimens (n chambers) | Total n of transect maps (n of selected transect maps) |
|---|---|---|---|
| A. tepida | 11 | 6(12) | 24(12) |
| B. marginata | 8 | 4(8) | 16(8) |
| A. lessonii | 8 | 6(9) | 16(9) |

*Table S1: Number of maps measured by EPMA per species, with number of specimens and chambers analyzed and total number of transect maps (S/Ca and Mg/Ca values reported in Fig. 6). Note that the number of chambers analyzed is higher than the number of maps, since some maps are selected on the cross sections of two chambers (see Fig. S1-3). The last column gives the*

***number of transects maps selected for peak-base analysis, which was limited to one per chamber to avoid overrepresenting of one chamber. The values of the peak base analysis are reported in Fig. 7 and Table 2).***

- Page 8, line 11. On line 7 the peak and base ratios are quoted as 56.5 and 10.2, which would equate to a ratio of 5.5, not 2.8.

*An error occurred in the text, this value should be 20.2 and not 10.2, as also stated in Table 2 and Fig. 7. This is now corrected*

- Page 8, lines 12-14. This is repetition of the first paragraph in this section.

*We reorganized this paragraph. We first described the co-variation of Mg and S distribution profiles per transect (example shown in Fig. 1C). Afterwards we compare the average S/Ca and Mg/Ca values of all transects, as shown in Fig. 6.*

- Page 8, line 20. Although the data are somewhat challenging to interpret, there are technically Mg/Ca-temperature data for Amphistegina reported by Raja et al. [2007] doi: 10.1029/2006GC001478.

*We are aware of this field study, and indeed, these results are not straightforward. However, we now added it to the text for completion.*

- Page 8, line 28. How is 0.9 derived? I calculate (35-20)*0.09 = 1.4 mmol/mol.

*We thank the reviewer for spotting this error, and it is changed accordingly.*

- Page 9, lines 1-2. How does this follow from the previous sentence? From Fig. 3 it appears that Mg and SO42- are not necessarily coupled, why do these data imply that they are?

*We rewrote this sentence: 'Since temperature-induced changes in Mg incorporation do not increase foraminiferal S/Ca, Mg/Ca and S/Ca might therefore co-vary due to a different process, possibly by mechanisms involved in biomineralization.'*

- Page 9, line 10. I don't think 'incorporated simultaneously' is the right terminology. Rather, the higher concentration bands are located in a similar place.

*We changed this to: 'are spatial (and hence likely temporally) correlated'*

- Page 10, lines 9-15. As it is written, it reads as if the mechanism for increased Mg/Ca resulting in increased alkali metal incorporation differs from that of the alkali earths. My understanding is that this is not necessarily the case, rather lattice distortion can result in increased incorporation into both lattice and interstitial sites (depending on ionic radius).

*We do not disagree that incorporation of sulfate could be influenced by lattice distortion in theory, but we would expect to see this in Fig. 3, and in the base-peak values between species, as noted at the end of the paragraph. However, we concur to remove 'alkali' to not limit the discussion.*

- Page 10, lines 23-27. I don't doubt that precipitation rate may be more sensitive to seawater Mg/Ca than temperature, but surely temperature will affect rate to some degree, if only because of the effect of temperature on carbon speciation through the temperature dependence of KW.

*We rephrased the end of this paragraph*

- Page 10, line 32. I suggest removing this sentence. There is no observational or theoretical evidence for inward-directed Mg transport, and it is difficult to see what the purpose of this would be.

*We changed this to 'passive transport or leakage of Mg'. Ca pump can accidently transport other ions than Ca. Due to (de)hydration of Mg, this ion would be less (or more) available for accidental transport, or leakage.*

- Page 11, lines 1-3. The (de)hydration of any ion during attachment is a passive process depending on e.g. growth rate and the chemistry of the calcifying space. Why is it only likely in one of the biomineralization models?

*Because in the case of SWV, you would expect the opposite, a decrease of Mg/Ca due to less efficient export of Mg.*

- Page 11, lines 12-15 and line 24. Given that MgCO3 is a small proportion of total Mg it seems unlikely that this is the explanation.

*This paragraph is rewritten in the revised version of the manuscript and de-emphasized presence of MgCO3 in the foraminiferal shell.*

- Page 11, lines 17-20. I don't follow the logic here. It reads as if the argument here is that the relationship between sulphate and temperature is counteracted either by the increased shell S/Ca being driven by the increased shell Mg/Ca, or that sulphur is actively transported to the calcification site to a greater degree at higher temperature. However, the first of these explanations is discounted elsewhere (e.g. page 14, line 12) and I do not see the mechanistic basis for the other based on the data presented here. To phrase it another way, surely the slope in Fig. 6 is being driven by the width or number of the co-located high-Mg, high-S bands in each transect, and unless the proportion of these changes as a function of temperature (does it?), why would the slope in Fig. 6 counteract a temperature-driven change in the activity of sulphate?

*We restructured this paragraph to make our arguments more clear. However, since the data set as such does not allow to investigate a linkage between banding and temperature, we removed lines 17-20. This is not affecting the overall structure and discussion.*

- Page 12, line 5. Evans et al. [2018] calculate that the Mg/Casw ratio at the calcification site of low-Mg foraminifera is <0.1 mol/mol, not 2 mol/mol as stated.

*We thank the reviewer for spotting this error. The values is now changed to <0.1*

- Page 12, line 27. Perhaps a bit picky, but it is better approximated to Ca/(CO32- +

SO42-) = 1.

*This is true, but then to be more precise, it also has to be Ca+Mg+Sr+Na etc. Since Ca and CO32- are the two main constituents, we leave the 'formula' as is for this exercise. We do change the 1 to ~1*

- Page 13, lines 16-18. I think this could be phrased more strongly, it is hard to see that the SO42-/CO32- ratio is less than one given ~30 mM [SO42-], unless the DIC concentration is very high in the calcifying space.

*We phrased this more strongly now, and added a sentence to this paragraph.*

- Page 13, line 22. It may well be species-specific, but note that the calcification site pH is not necessarily greater than 9 [~8.75 according to Bentov et al., 2009].

*We added the value of Bentov et al., 2009 to this line, to show variability between species (or methods).*

- Page 24, lines 24-28. In both biomineralization models, pH is elevated in the calcifying space (or vacuole) in order to promote carbon concentration, and presumably the two processes occur simultaneously (what would be the benefit of separating them?). We don't know precisely to what

extent this takes place or at what time, so I understand the reason for calculating it in this way, however my recommendation would be to rephrase this sentence as a constraint on the maximum calcification site SO42-/CO32- ratio, given that the assumption of seawater DIC is probably not correct.

*We added some sentences to clearly state this would be a maximum value, which will in the end depend on DIC. However, it is unlikely that DIC will increase to a point where SO42:CO32 will be <1, and we added a short sentence in the revised version of our manuscript.*

- Page 14, line 13. I don't think anyone has suggested that hyaline and porcelaneous foraminifera are characterised by the same biomineralization model.

*We rephrased this sentence.*

- Fig. 1. State which species/treatment this map is from.

*OK*

- Fig. 3. Please clarify whether the grey symbols represent repeat measurement of the same solution or different groups of 10 foraminifera.

*We changed the caption, it now states that these are not repeat measurements but different groups.*

- Fig 9. Proton pumping is a feature of both biomineralization models, so why should the arrow for 'SWV-dominated' be in the opposite direction to 'H+ pump-dominated'?

*See comments above. We acknowledge that SWV also includes a small increase in pH (8.75 according to Bentov et al., 2009). However, the pH increase in the vacuoles is lower then observed inside small benthic foraminifera. Therefore, we reduced the size of the arrow, but kept the direction as it was.*

- Tab. 1 could be moved to the supplement.

*We decided to keep table 1 in the main manuscript, since it is an important part of the LA ICP MS method.*

Typos

- Page 2, line 31. There is a full stop missing after the parenthesis.

*Done*

- Page 5, lines 28 and 33. Presumably it should read µl/min.

*Done*

- Page 6, line 13. An emulsion refers to two immiscible liquids, replace with suspension.

*Done*

- Page 7, line 23. Replace 'is' with 'are'.

*Done*

*Anonymous reviewer #2,*

*Thank you for the constructive comments we received on our manuscript. Below, all our comments are listed in italics. We try to answer all specific comments raised by your review.*

The submitted text provides interesting and intriguing new data. The methods are appropriate, the results require more descriptions. Data are not quite properly presented and errors not properly discussed. Individual error bars are not shown in the figures nor discussed in the text. Yet, for example, there is an error both and the X and Y axes in figure 3. In the results section, only the error on the averages is provided. What is the instrumental reproducibility? The external reproducibility?

*We added error bars in Figure 3a and b. Horizontal error bars represent the standard deviation of the temperature conditions during the experiment, and vertical error bars indicate the variability based between measurements (which were done on different samples). This information was also added to the figure caption.*

*The instrumental reproducibility for the elemental data in Figure 3 is much smaller, 0.4% and 1.0-1.7% for Mg/Ca and S/Ca respectively (this will now be added to section 2.3.3.). Propagation of the analytical error (internal precision) is negligible compared to the external error, which we hence indicated by the error bars. Comparison between internal and external precision is added to section 2.3.3. The errors themselves will be listed in section 2.3.*

*In manuscript: Accuracy of Mg/Ca is 105% and 101% for JCt-1 and JCp-1, respectively with an external precision of 0.4% for both standards. Only JCp-1 has a certified value for S/Ca, and accuracy for our measurements is 94% based on this standard. The external precision of S/Ca is 1.7% and 1.0% for JCt-1 and JCp-1.*

Improvements can be made regarding the writing (I provide a few suggestions in the detailed comments). The path leading to the conclusion that Mg and S correlate at the bulk scale (EPMA profiles) could be improved. Obviously, even if there is a Mg-S correlation, the fact that there is a Mg-T and no S-T correlation is intriguing. Based on the data provided, the correlation could be described as significant for B. marginata but not for the other species analyzed here. Furthermore, when the authors present a correlation between EPMA-based Mg/Ca and S/Ca data, the values given in the text (page 7) do not match the values given in the caption of figure 6.

*This section was apparently not very clear. We showed both correlations (coefficients and p-values) within transects (section 3.3) and between transects (figure 6). For clarity we will remove the correlations within the transects as we do not further use these here. An example is still given in figure 1c. The coefficients of determination and p-values (more than 95% significant) of the relation between S/Ca and Mg/Ca between transects were stated in the caption of Fig. 6, and we now also will add them to the main text (section 3.3).*

Some questions could be addressed more explicitly.

1) do the EPMA correlation represent each one individual or more?

*Based on the confusion raised by this section of both reviewers, we added a section to the revised version of the manuscript describing which analyses come from where (species, specimens,*

*chambers) and in table S1 we list the S/Ca and Mg/Ca values plotted in Fig. 6, which are averages on transects. Hence, every point plotted in figure 6 and value listed in table S1 is based on averaging an individual transect. Figure 6 includes sometimes more than one transect per map (i.e specimen), which is now added to table S1. The location of the maps and the number of transects per map, specimens and species can now also be found in Figures S1-S3.*

2) the individual errors should be reported on the figure, and the authors need to provide the MSWD. Are the correlations still "significant" when individual errors are taken into account?

*We added the standard errors of the individual points to Fig. 6. This does not necessarily help to better show significance of the observed correlation, but at least make it visually easier to appreciate the correlations. As every symbol presents the average E/Ca of a transect map and the number of analyses included is rather high, the statistical power of the here presented data set is quite high too. Even when plotting all raw data the inferred correlations remain clear. We added this figure in the rebuttal for the reviewer (Fig. R2: All data points from all EPMA transects per species.).*

3) the authors should define what they consider as "significant". Is there enough points for the p-value to be meaningful?

*The p-value is based on both the calculated t-value and the number of points (i.e. degrees of freedom, which is based on the number of points minus one). The p-values here reported are very low, <0.0005 and 0.0025, indicating that the confidence limits for the correlations are higher than 99%. This was mentioned in the caption only, but will be also added to the main text for clarity (section 3.3.)*

4) The forams used for those maps were grown under hypoxia. Can it alter the comparison? Why not use forams grown in regular oxic conditions similarly to those used in this manuscript? It seems to me that it would have made more sense to use for instance individuals from the temperature experiments. What would the EPMA correlation look like if we took specimens from Fig. 3? That could help unravel the Mg-S, Mg-T correlation and lack of S-T correlations. I feel that the maps don't quite fit in the current story, and I wonder if that could be due to the fact that they were performed on specimens from other experiments.

*The foraminifera grown under controlled temperature conditions were dissolved and measured for S/Ca on SF-ICP-MS, leaving no material for high-resolution study by EPMA. The advantage of the SF-ICP-MS analyses is that these analyses are much more precise and quantitative in contrast to the semi-quantitative analyses from EPMA. However, for these traditional solution analyses more material is needed, which is often sparse from culture experiments. Therefore we used specimens available from another experiment to analyse S/Ca – Mg/Ca at the inter chamber level by EPMA. Therefore, EPMA was performed on specimens of low oxygen experiments, which were already planned to be investigated on Mn/Ca, with culturing at 30% oxygen. These specimens are in our opinion also suited for the S/Ca study as foraminifera are known to be able to endure hypoxia and even (short) term anoxia, in both culture studies and in natural environments. These experiments*

*never went below 30% and hence did most likely not alter calcification as is also evident from the large number of chambers added. Therefore, we assume uptake and incorporation of elements occurred normally. We will add some sentences to make readers aware of the fact that we used results from multiple experiments.*

Here follow some questions and thoughts about the interpretations the authors make of their data.

-Why show only MgCO3 and not Mg only activity in figure 8? Based on section 4.3.1, the MgCO3 theory does not seem strongly supported by anything else than the current PHREEQC calculations. In the absence of reference further supporting the idea cited in the text, it should be much more developed). It would be more relevant to show all species as simple ions AND CaCO3 and MgCO3 to al allow the reader to make their own opinion. I know that the atuhors can't plot every single dissolved species, but what they show seems too partial.

*In the text we state that the activity of Ca and Mg remain stable in this case over the studied range and that we therefore do not plot these. The reason for this is that the free Mg2+ has a much higher abundancy, not affected appreciably by the changes in pH and/or temperature. The more rare species, however, are affected. Hence, we show only activities of these species, selected based on their sensitivity. This is also needed to keep the figure readable, as certain species have activities differing more than a factor of thousand. We now state these differences more clearly in the text (4.3.1.).*

-L. 33, the authors state that "an increased removal of Mg at higher temperatures would results in a lower Mg/Ca at higher temperature". That sounds logical as such, but they omit to explain why removal of Mg should increase at higher temperature in the first place.

*This was added to discussion to explain the difference between inward and outward transport of cations during biomineralization. At higher temperature, Mg removal would increase, since at higher temperature there would be more dehydrated and therefore transportable Mg, which is now stated more clearly in the text (section 4.3.1.). The conclusion of this paragraph is that the temperature dependent transport is in line with TMT (inward bound cations) and not with SWV (outward bound cations).*

*In manuscript: "Since dehydration of magnesium ions costs less energy at higher temperatures, it may be expected that there would be more dehydrated and transportable Mg available. This would lead to an increased (accidental) transport of Mg2+ to the SOC by Ca2+-pumps leading to a positive effect of temperature on Mg/Ca, or an increased selective removal of Mg2+ resulting in theory in a lower shell Mg/Ca at higher temperatures."*

-Similarly, some possible mechanisms of correlation of Mg and S are not discussed. Tanaka et al., 2018 provide a thorough review of some mechanisms not considered here that are not to affect Mg in abiogenic calcite, such as the role played by organic matter, growth rate, saturation state rayleigh effect, or even modification of the calcifying fluid composition. The authors splits the processes as purely inorganic or biomineralization process-related (or a mixture of both) and do not include, for instance, the effects of organic matter on calcification. S is (too) often used as a mere way to track organic matter in the shell, without further discussion, but here this option is not even discussed. In section 4.2 the authors mention that S and Mg follow organic linings but then they do not follow up on that question. Yet, the presence of some type of OM can affect Mg (effect on SO4 unknown) content (Mavromatis et al., 2017) and could provide a source of correlation.

*Due to the large number of processes potentially involved in (inorganic) precipitation, we decided to somewhat limit the number of mechanisms discussed to the here most likely relevant. In a previous*

*analysis of S distribution in Amphistegina (van Dijk et al., 2017) we observed an offset between organic linings and elevated S/Ca bands. Even though it is still possible that higher amounts of organic material of the POS somewhat increase Mg/Ca and S/Ca the work of Busenburg and Plummer 1985 and Kitano et al., 1975, show that $SO_4^{2-}$ (as well as Na, Amiel et al., 1973) is predominately present in biogenic calcite in solid solution and not a component of the organic matrix. We will add this mechanism to the beginning of section 4.2 (see below).*

*In manuscript: "The presence of organic material could cause a higher Mg content due to increased adsorption of Mg (Mavromatis et al., 2017). If also the case for other elements, including S, this could explain the observed covariation within chambers (Fig. 5), as earlier suggest by (Kunioka et al., 2006). However, this is disputed by the work of Busenburg and Plummer (1985) and Kitano et al. (1975), which shows that $SO_4^{2-}$ (as well as Na, Amiel et al., 1973) is predominately present in solid solution and not as a component of the organic matrix of biogenic (Mg-)calcites."*

-The authors could mention the SO4/Ca Kd evaluated by Kitano et al., (1975) or Busenberg and Plummer (1985) in their figure 9 or in the discussion.

*Partition coefficient of SO42- ($D_{x1000}$) by Busenberg and Plummer 0,013-0,774 (synthetic calcites) is now added to the discussion section.*

-P. 12,L.5, L. 16 : why assume that sulfate is constant? That seems unlikely as we know that calcite cannot precipitate at seawater sulfate concentrations. (This point is partially adressed in section 4.3.4, so it would probably make more sense to move up that section). In addition, the vacuole pH is modified (Bentov et al., 2009), so the SO4/CO3 ratio would necessarily be modified. Therefore this process needs to be reassessed. The Mg/Ca ratio might be lowered (not "Mg" as stated line 16), but the SO4/CO3 ratio will certainly be modified as well regardless of what happens to sulfate.

*We thank the reviewer for the suggestion and agree that within the SOC the SO4/CO3 is modified. Therefore, we changed the vector, since due to an observed increase of pH in the vacuoles, CO32- for sure would be increased, and therefore SO4/CO3 decreases. Similarly, a reduction in SO4 would be theoretically helpful in biomineralization as well, although evidence for this is still lacking. Accordingly, in Fig. 9 the vector is now decreased in size, albeit that we kept the original orientation. We now also refer to Bentov et al., 2009 and mention an increase in vacuole pH in section xx.*

*In manuscript: "In the seawater vacuolization (SWV) model (Bentov et al., 2009), the main source of ions is from the endocytosis of seawater. The Mg/Ca of the fluid in these seawater vacuoles is lowered (<0.1 mol/mol; Evans et al., 2018), but it is not known if the sulfate concentration is regulated in these vacuoles, making it impossible to assess whether $Mg^{2+}$ and $SO_4^{2-}$ concentrations in the vacuoles are correlated. However, the (small) increase in pH of the vacuoles (~8.7 for species Amphistegina lobifera; Bentov et al., 2009) can decrease the $[SO_4^{2-}]/[CO_3^{2-}]$ of the vacuoles."*

and

*"i)     SWV dominated: During endocytosis, Mg/Ca in the vacuoles will be actively lowered, while $[SO_4^{2-}]/[CO_3^{2-}]$ in the vacuoles is lowered due to increase of pH in the vacuoles."*

*Figure 9 revised:*

[Figure]

*Figure 9: Average values of S/Ca, expressed as $SO_4^{2-}/CO_3^{2-}$, versus Mg/Ca of different hyaline (circles) and porcelaneous (diamonds) species of foraminifera of this study (open symbols) and other studies (grey symbols), including values obtained for the NFHS (for a description see Mezger et al., 2016). Culture experiments with varying salinity (c), temperature (1) and $pCO_2$ (d,e), min and maximum ranges are indicated by grey bars, but in most cases fall within the symbol. Values of foraminiferal E/Ca can be found in Table S4. Values for Mg/Ca of inorganic precipitated calcite of Mucci and Morse, 1983 are indicated by the gray bar, SD is included. Note the inversed axis compared to Fig. 4.*

-P. 12 L.17: I also disagree with this vector. Is it possible to Ca++ to cross a membrane and have no proton exchange?

*It is not possible for charged ions to cross a membrane without an equally charged counter current (or simultaneous current of opposite charge). Form many systems, it is shown that protons are co-transported with Ca2+. Hence, we maintained the vector as is, and use it merely to point out the general direction based on the constrains given by this specific biocalcification model.*

-P. 13 L.15-16: my understanding of Fernandez-Diaz (2010)'s paper is different. In this study, calcite+vaterite precipitates no matter what the SO4/CO3 ratio is. However, vaterite needs not be stable (see Jacobs et al. 2017: vaterite evolves to calcite by dissolution-reprecipitation). Fernandez-diaz and collaborators state that vaterite "transform into the more stable calcite via dissolution-reprecipitation although such transformation is hindered when the SO4/CO3 ratio in the fluid is higher than 1.3". This would put a higher limit on the sulfate content of the fluid in which the dissolution reprecipitation needs to happen, maybe not necessarily a lower limit as suggested by the authors of the current manuscript.

*We changed this sentence accordingly: In manuscript: "Vaterite transform into calcite via dissolution-reprecipitation when solution $SO_4^{2-}:CO_3^{2-}$ < 1.3 (Fernández-Díaz et al., 2010)."*

As a result, it seems that figure 9 needs more constraints in order to make a solid case. The questions that are not solved by that figure are:

-models often attempt to explain E/Ca ratios and therefore don't consider the fate or role of protons (eg. Langer et al., 2006; Nehrke et al., 2013) and sulfate is almost never considered. Combining them to explain Mg/Ca vs. SO4/CO3 would therefore more thinking about the role played by protons, a common way to ensure electroneutrality across a membrane when pumping or channeling ions or what happens to sulfate therefore seems possibly problematic.

*Theoretically Mg2+ could exchange for SO42- in order to maintain electroneutrality. However, the fact that we here observe a positive correlation on all scales effectively rules out such a mechanism. This implies, as the reviewer indicates, that proton pumping is more likely involved in pumping/ion channelling.*

-does it work also for the EPMA profile-scale correlations? This scale does not seem to be interpreted in the text

*The variation, or banding, observed in the EPMA maps are not due to a mixing of processes in biocalcification. It is due to a co regulation of Mg/Ca and SO42/CO32 at the site of calcification. In Fig. 9 we tried to explain the Mg/Ca and S/Ca of two groups of foraminifera in terms of calcification mechanisms, TMT versus SWV. We now stated this more clearly in the text.*

-does it make sense to calculate a R2 in figure 9? In the end, it is almost like a linear correlation between two (group of) points...

*We agree with the reviewer and remove the R2 in Fig. 9.*

-if the authors invoke vaterite as the precipitating polymoprh to solve the problem of sulfate inhibition on calcite precipitation, why use inorganic calcite in this figure?

*We would like to refrain from invoking vaterite precipitation to explain the sulfate inhibition. However, as this recent discovery does have potential implications for biomineralization models we still included this here. Partitioning of Mg in vaterite is not studied in such a way that it allow comparison to foraminiferal calcification, as also mentioned by Jacob et al., 2017. We hence give the inorganic Mg/Ca values for calcite, also to facilitate comparison to previous studies. This is now explained in section 4.3.2.*

*In manuscript: "Furthermore, in Fig. 9 we also present Mg/Ca values of inorganic calcite from Mucci and Morse (1983), values that are often used to compare Mg/Ca values of foraminifera with inorganic calcite (Evans et al., 2015; van Dijk et al., 2017). However, new evidence has arisen that foraminifera might precipitate vaterite, which ultimately transforms to calcite, indicating a complex pathway and partitioning of elements during calcification."*

Detailed comments

P. 1 L. 14 substitutes for (and later in the text)

*Done.*

L. 16 we analyzed the bulk concentration (overall in the text it would be good to ensure lack of ambiguity between bulk and in-situ analyses)

*We changed the terminology in the revised version of the manuscript to avoid confusion about bulk and high resolution in situ analyses.*

L. 24 consistent small-scale covariation (same comment as above)

*See answer above.*

P.2 L.7 substitutes for

*Done.*

L. 31 no "." before Removal

*Done.*

P.3 L.3 do the authors really investigate the co-varaition between Mg content and S incorporation? It seems to me that they investigate the co-variation between Mg and S contents to provide new constraint on biocalcification. The sentence should be improved.

*Changed.*

*In manuscript: "Here we investigate the co-variation between magnesium and sulfur content of different species of foraminifera to provide new constrains on biomineralization."*

P.4 L.5 temperature errors should be reported on fig. 3

*We added the SD of the temperatures of the different treatments in Fig. 3:*

[Figure]

Section 2.3: check exponents in isotopes. The name of the instruments are lacking. The laser is provided, not the various MS or EP. what are the blanks? the instrumental backgrounds? The instrumental error? The reproducibility of the data? The whole section needs more solid writing on the data quality and statistics.

*We added the specs of the SF ICP MS, as well as the precision and accuracy (reproducibility), as mentioned above. The section is updated to contain the requested parameters. Exponents of the isotopes are now in superscript.*

Section 2.3.3 signification of SF-ICP-MS not provided (could be done line 10)

We added the definition of the SF ICP MS.

P. 7 L. 23-28: the R values do not match that of figure 7 and the errors provided are not homogeneous. The p-values are just mentioned in the caption but no explanation is provided in the text. The notion of significance should be defined.

*The coefficients of determination and p-values of the relation between S/Ca and Mg/Ca were stated in the caption of Fig. 6, but we now also state them in the main text now. The previous p values were based on the $R^2$ of the profiles (for example see 1C). However, we replaced these now for the coefficient of Fig. 6, to avoid confusion.*

P.9 L.7 it seems worth referring here to Paris et al. (2014) nano-SIMS data as well, which do show that with much higher precision, the bands are not quite identical, at least in O. universa.

*We now also refer to Paris et al., 2014.*

P. 10 L. 17 is the temperature-induced increase of Mg due to crystal lattice distortion change? It has too, in order to use this as an argument here, but it does not seem to be the case based on the explanations provided in the text.

*We removed the word solely here, to avoid any confusion.*

P. 11 L.26 The conclusion sentence should be rewritten. It seems strange to write down that it cannot be excluded, when the authors themselves are the one suggesting it.

*We rephrased this sentence.*

*In the manuscript: Hence, changes in the amount of MgCO$_3$ complexes does not explain the full range observed*

Figure 9 caption: L.2 species of foraminifera?

Added.

**Coupled Ca and inorganic carbon uptake suggested by magnesium and sulfur incorporation in foraminiferal calcite**

Inge van Dijk[1,2], Christine Barras[1], Lennart Jan de Nooijer[2], Aurélia Mouret[1], Esmee Geerken[2], Shai Oron[3], Gert-Jan. Reichart[1,4]

[1]LPG UMR CNRS 6112, University of Angers, UFR Sciences, 2 bd Lavoisier 49045, Angers Cedex 01, France.
[2]NIOZ Royal Institute for Sea Research, Department of Ocean Systems (OCS), and Utrecht University, Postbus 59, 1790 AB Den Burg, The Netherlands.
[3]Charney School of Marine Sciences, University of Haifa, Israel.
[4]Utrecht University, Faculty of Geosciences, Budapestlaan 4, 3584 CD Utrecht, The Netherlands.

*Correspondence to*: Inge van Dijk (Inge.van.Dijk@nioz.nl)

**Abstract.** Shell chemistry of foraminiferal carbonate proves to be useful in reconstructing past ocean conditions. A new addition to the proxy toolbox is the ratio of sulfur (S) to calcium (Ca) in foraminiferal shells, reflecting the ratio of $SO_4^{2-}$ to $CO_3^{2-}$ in seawater. When comparing species, the amount of $SO_4^{2-}$ incorporated, and therefore the S/Ca of the shell, increases with increasing magnesium (Mg) content. The uptake of $SO_4^{2-}$ in foraminiferal calcite is likely  connected to carbon uptake, while the incorporation of Mg is more likely related to Ca uptake since this element substitutes for Ca in the crystal lattice. The relation between S and Mg incorporation in foraminiferal calcite therefore offers the opportunity to investigate the timing of processes involved in Ca and carbon uptake. To understand how foraminiferal S/Ca is related to Mg/Ca, we analyzed the concentration and within-shell distribution of S/Ca of three benthic species with different shell chemistry: *Ammonia tepida*, *Bulimina marginata* and *Amphistegina lessonii*. Furthermore, we investigated the link between Mg/Ca and S/Ca across species and the potential influence of temperature on foraminiferal S/Ca. We observed that S/Ca is positively correlated with Mg/Ca on microscale within specimens, as well as between and within species. In contrast, when shell Mg/Ca increases with temperature, foraminiferal S/Ca values remain similar. We evaluate our findings in the light of previously proposed biomineralization models and abiological processes involved during calcite precipitation. Although all kinds of processes, including crystal lattice distortion and element speciation at the site of calcification, may contribute to changes in either the amount of S  or Mg that is ultimately incorporated in foraminiferal calcite, these processes do not explain the  co-variation between Mg/Ca and S/Ca values within specimens and between species. We observe that groups of foraminifera with different calcification pathways, e.g. hyaline versus porcelaneous species, show characteristic values for S/Ca and Mg/Ca, which might be linked to a different calcium and carbon uptake mechanism in porcelaneous and hyaline foraminifera. Whereas Mg incorporation is might be controlled by Ca dilution at the site of calcification due to Ca-pumping, S is linked to carbonate ion concentration via proton pumping. The fact that we observe a coregulation of S and Mg, within specimens and between species suggests that proton pumping and Ca pumping are intrinsically coupled across scales.

**1. Introduction**

The elemental and isotopic composition of foraminiferal calcium carbonate shells reflect seawater chemistry, and is therefore widely used to reconstruct specific marine environmental conditions. Besides the potential of Mg/Ca and $\delta^{18}O$ to reconstruct seawater temperature, currently available proxies permit reconstruction of part of the marine inorganic carbon system (Beerling and Royer, 2011; Hönisch and Hemming, 2005). One of the most recent additions to the proxy tool box is the sulfur to calcium ratio (S/Ca) values of foraminiferal shells. In both abiogenic and biogenic carbonates, sulfur is mainly present in the form of $SO_4^{2-}$, where it substitutes for $CO_3^{2-}$ (Pingitore et al., 1995; Perrin et al., 2017). S/Ca is correlated to the ratio of $SO_4^{2-}$ and $CO_3^{2-}$ in seawater in both inorganic carbonates (Fernández-Díaz et al., 2010) as well as in foraminiferal calcite (Paris et al., 2014; van Dijk et al., 2017a). However, the few calibrations on foraminifera currently available are for the species *Amphistegina gibbosa* and *Sorites marginalis* and show species-specific offsets: the amount of $SO_4^{2-}$ incorporated, and therefore the S/Ca, increases with increasing Mg content (van Dijk et al., 2017a).  Covariation of concentrations of S and Mg across species could be due to  increased incorporation of $SO_4^{2-}$ over $CO_3^{2-}$ as a response to elevated crystal lattice strain due to higher concentrations of other elements, like Mg (Mucci and Morse, 1983; Evans et al., 2015), or 2). Since the ionic radius of $SO_4^{2-}$ is larger than $CO_3^{2-}$, it might indeed be possible that distortion of the lattice by Mg leads to substitution of $CO_3^{2-}$ by $SO_4^{2-}$. Another explanation would be  co-
[revised manuscript text omitted]
$^2$). We choose MACS-3 as a calibration standard, since element composition approaches the foraminiferal values closer than that of NIST 610 or 612 and therefore aids a more robust

10   calibration, even though the MACS-3 is slightly less homogeneous (see precisions listed in Table 1). Accuracy and precision per element per standard are reported in Table 1.

In total, 441 chambers were measured; 142 ablations on 59 specimens for a temperature of 21.2°C, 189 ablations on 63 specimens for 26.3°C and 110 ablations on 42 specimens for T=29.5°C. Element concentrations were calculated by integrating individual laser-ablation profiles using an adapted version of the data reduction software SILLS (Signal

15   Integration for Laboratory Laser Systems; Guillong et al., 2008) package for MATLAB (Geerken et al., 2018; van Dijk et al., 2017b). Profiles were selected to avoid contamination of the outer or inner part of the foraminifera (for examples of profile selection, see e.g. Dueñas-Bohórquez et al., 2011; Mewes et al., 2014; van Dijk et al., 2017c). Average E/Ca per temperature conditions were calculated after removal of outliers (based on 1.5*Interquartile range). We applied a t-test to assess if E/Ca is different between temperature conditions using a bilateral test.

20   **2.3.3. Bulk measurements by SF-ICP-MS**

Grouped foraminifera from the controlled temperature experiment and the species from Burgers' Zoo stock (*Amphistegina lessonii, Heterostegina depressa, Sorites orbiculus*, *Spiroculina angulata*, *Spiroloculina communis*, *Quinqueloculina pseudoreticulata* and *Quinqueloculina* sp.) were analyzed for the sulfur content in their shells. Foraminifera from the controlled temperature experiment received an additional cleaning step (following the same procedure as van Dijk et al.,

25   2017a), since they were previously fixed on a laser ablation stub with tape. These specimens were transferred in groups of ~approximately ten individuals to 0.5 ml acid-cleaned TreffLab PCR-tubes and rinsed three times with 750 μl of de-ionized water, during which the vials were transferred to the ultrasonic bath for 1 minute (80 kHz, 50% power, degas function). Subsequently, the samples were again placed in the ultrasonic bath for another minute after addition of 750 μl of suprapur methanol (Aristar). The solvent was removed and the samples were rinsed three times with ultrapure water. Vials were dried

30   in a laminar flow cabinet.

Samples from the temperature experiment and from the Burgers' Zoo were measured on different occasions, using a slightly different analytical approach. Both groups of samples were measured on an Element-2 (Thermo scientific) sector field double focusing inductively coupled mass spectrometer (SF-ICP-MS). For the samples of the temperature experiment, 150

µl of ultrapure 0.05 M $HNO_3$ (PlasmaPURE) was added to each vial to dissolve all foraminiferal calcite. A five second pre-scan for $^{43}Ca$ was performed on the  SF-ICP-MS to determine the [Ca] in the dissolved foraminiferal calcite solutions and accordingly to these results, samples were diluted to obtain a solution with 40 ppm Ca. After the cones were preconditioned during 2 hours with 40 ppm pure $CaCO_3$, final solutions were measured again for ~170 seconds per sample. Samples were injected into the ICPMS using a microFAST MC system of ESI with a loop of 250 mm and a flow rate of 50 ml/min. For the sample set of the temperature experiment, masses $^{23}Na$, $^{24}Mg$, $^{32}S$, $^{34}S$ and $^{43}Ca$ were analyzed in medium resolution to separate $^{16}O^{16}O$ from the $^{32}S$ peaks, and $^{18}O^{16}O$ from the $^{34}S$ peaks.

The samples from Burgers' Zoo were dissolved in 0.5 ml 0.1 M $HNO_3$ and diluted to 100 ppm Ca accordingly to the results of the Ca pre-scan. Elemental composition of the foraminifera was measured for a wide range of elements, including $^{23}Na$, $^{24}Mg$, $^{32}S$, $^{34}S$ and $^{43}Ca$ at medium resolution. In total 46 isotopes were measured during six min at low, four min at medium and one minute at high resolution with a 300 ml/min flowrate using a peristaltic pump.

For both sets of measurements, samples were measured against six ratio calibration standards with a  matching matrix, i.e. 40 ppm Ca for the temperature set and 100 ppm Ca for the Burgers' Zoo set.  In addition to the foraminiferal samples, we measured several standards, including NFHS-1 (NIOZ Foraminifera House Standard; for details see Mezger et al., 2016), JCt-1 (Giant Clam, *Tridacna gigas*) and JCp-1 (coral, Porites sp.; Okai et al., 2002) to monitor drift and the quality of the analyses. One of the ratio calibration standard was measured after every 5th sample to monitor drift. Accuracy of Mg/Ca is 105% and 101% for JCt-1 and JCp-1, respectively with an external precision of 0.4% for both standards. Only JCp-1 has a certified value for S/Ca, and accuracy for our measurements is 94% based on this standard. The external precision of S/Ca is 1.7% and 1.0% for JCt-1 and JCp-1. We used a ratio calibration method (de Villiers et al., 2002) to calculate foraminiferal S/Ca (mmol/mol).

**2.3.4.  Shell wall variability by EPMA**

Specimens of various foraminiferal species (*Ammonia tepida*, *Bulimina marginata*) from a recently published culture study (Barras et al., 2018), and *Amphistegina lessonii* cultured in the same culture set-up,  were prepared for electron probe micro-analysis (EPMA) to investigate the intra-shell incorporation of sulfur and magnesium. These foraminifera were cultured under hypoxia (30% oxygen saturation) in controlled stable conditions and previously studied to investigate the Mn incorporation in foraminiferal calcite (for details and culture methodology, see Barras et al., 2018). *Ammonia tepida* and *Bulimina marginata* were cultured at 12°C, while specimens of *Amphistegina lessonii* were grown at 25°C. For the latter species, the set-up was equipped with a light system with 12 hr/12 hr light cycle."  Specimens of each species were embedded under vacuum in resin (2020 Araldite® resin by Huntsman International LLC) using 2.5 cm epoxy plugs. Samples were polished using increasingly finer sanding paper. In the final polishing step, a diamond  suspension with grains of 0.04 µm was used, resulting in exposure of cross section of chamber walls. After applying a carbon-coating, the samples were placed in the microprobe sample holder. After selection of

target areas, several small high-resolution maps (130x97 pixels) were analyzed at 12.0 kV in beam scan mode for different elements (Ca, Mn, Mg, S and Na) with a dwell time of 350 ms. In total, we analyzed  12, 8 and 9 chambers of  6, 4 and 6 specimens of *Ammonia tepida*, *Bulimina marginata* and *Amphistegina lessonii*, respectively., an overview of the number of maps per species and the location of the maps and transects can be found in the supplementary information, Table S1 and in Fig S1-S3.

[revised manuscript text omitted]

**4.3.1. Calcite precipitation and physico-chemical conditions**

The observed link between S/Ca and Mg/Ca might be explained by investigating parameters involved in inorganic precipitation studies. Chemical processes operating at the crystal-solution interface or in the fluid contained in the site of calcification might give insights in the observed correlation between sulfate and magnesium incorporation in foraminiferal calcite, as well as the temperature effect on Mg incorporation. Magnesium ions in the parent solution have been found to increase the co-precipitation of other elements (Okumura and Kitano, 1986). However, this is observed for alkali metal ions, which are in interstitial positions or substitute for $Ca^{2+}$ in the crystal lattice, while $SO_4^{2-}$ is hypothesized to exchange for $CO_3^{2-}$ ions (Pingitore et al., 1995; Perrin et al., 2017; Berry, 1998). Besides promoting co-precipitation, incorporation of magnesium in carbonate is suggested to cause strain on the crystal lattice, leading to distortion, and an increase in the incorporation of other  elements (Mucci and Morse, 1983). This theory has been used to explain incorporation of certain elements, like $Na^+$ and $Sr^{2+}$ in larger benthic foraminifera (Evans et al., 2015). Besides the lack of study on the incorporation of sulfate with increasing Mg content, based on our data, the correlation between Mg/Ca and S/Ca cannot be explained  by crystal lattice distortion. The lack of response of S/Ca to (temperature-induced) changes in Mg/Ca (Fig. 3) together with the similar base values of S/Ca for all three investigated species, while base values for Mg/Ca vary between species (Fig. 7), show that S/Ca and Mg/Ca are not always correlated, which should be the case with this hypothesis.

The effect of temperature on Mg/Ca has been comprehensively studied in inorganic carbonate, by controlled precipitation experiments (for a summary see Mucci, 1987). The last decades, several explanations have been proposed to explain the relation between temperature and Mg incorporation in inorganic carbonates. Firstly, the partitioning of certain elements in

inorganic experiments heavily relies on precipitation rate (e.g. Lorens, 1981). This was suggested for Mg incorporation (Chilingar, 1962), which may indicate that the increase of foraminiferal Mg/Ca in our study could be explained by a positive effect of temperature on precipitation rate. However, this was disputed by several studies (e.g. Mucci and Morse, 1983; Mucci et al., 1985), showing precipitation rate does not change with temperature, but is depending on the Mg/Ca ratio of the parent solution. However, to test if Mg/Ca increases, we would need to study the precipitation rate of foraminiferal calcite as a function of temperature. However, based on study on synthetic calcite, partitioning of $SO_4^{2-}$ would also increase with precipitation rate (Busenberg and Niel Plummer, 1985), which we do not observe in our study (Fig. 3).

It has been proposed that calcium- and magnesium transport to the site of calcification requires complete or partial dehydration of these ions, an energy-consuming process that is influenced by temperature (Mucci, 1986; Morse et al., 2007; Arvidson and Mackenzie, 2000). Re-hydration of these ions at the site of calcification (SOC) may furthermore determine isotopic fractionation during calcium carbonate precipitation (Mavromatis et al., 2013). Since dehydration of magnesium ions costs less energy at higher temperatures, it may be expected that there would be more dehydrated and transportable Mg available. This would lead to an increased (accidental) transport of $Mg^{2+}$ to the SOC by $Ca^{2+}$-pumps leading to a positive effect of temperature on Mg/Ca, or an increased selective removal of $Mg^{2+}$ resulting in theory in a lower shell Mg/Ca at higher temperatures. Since the latter is not observed, the effect of the (de)hydration of Mg ions is only likely in biomineralization models where Mg is not selectively removed, like the Trans Membrane Transport (TMT) mixing model (Nehrke et al., 2013). Although this explains the lack of a clear temperature effect on S/Ca-values, it does not explain coupled the coregulation of S/Ca and Mg/Ca behavior.

[revised manuscript text omitted]

foraminifera might precipitate vaterite, which ultimately transforms to calcite, indicating a complex pathway and partitioning of elements during calcification.

**4.3.34. Sulfate at the site of calcification**

Species specific differences in the relative contribution of SWV and TMT might provide an explanation of our results. However, while this could give insights in the incorporation of S and Mg as a function of temperature and explain species-specific differences, we did not consider the inhibition effect of sulfate and the probable non-classical calcification pathway foraminifera utilize to create their shell (Jacob et al., 2017). Sulfate is a known inhibitor for precipitation of calcite (e.g. Manoli and Dalas, 2000; Kitano, 1962), but does play a role in the transformation from amorphous calcium carbonate into vaterite (see Bots et al., 2012 and references therein). and stabilizes this metastable carbonate phase vaterite Vaterite transform into calcite via dissolution-repreipitation wwhen solution $SO_4^{2-}$:$CO_3^{2-}$ < 1.3 (Fernández-Díaz et al., 2010). A recent study has proposed that certain species of planktonic foraminifera create their shell by a pathway involving vaterite phases that transform ultimately to calcite (Jacob et al., 2017), which might suggest the $SO_4^{2-}$:$CO_3^{2-}$ at the site of calcification is >-1 when precipitation of the carbonate shell commences. Just prior to formation of a new chamber, the sulfate concentration at the SOC is probably similar to that in seawater, assuming the calcification fluid is composed of either a small volume of seawater enclosed by the protective envelop separating the SOC from seawater or by seawater vacuoles (SWV model). With a seawater concentration of ~30 mM [$SO_4^{2-}$], it is very likely $SO_4^{2-}$:$CO_3^{2-}$ at the site of calcification is > 1, but it is depending of the carbonate chemistry at the SOC.

Laboratory experiments have revealed that the internal pH of a foraminifera is elevated to, (species-specific) values ranging from ~8.75 (Bentov et al., 2009) to ≥9 (de Nooijer et al., 2009) at the start of shell formation (de Nooijer et al., 2009) due to proton pumping (Toyofuku et al., 2017), which lowers the pH in the microenvironment surrounding the foraminifer (Glas et al., 2012). When assuming the $SO_4^{2-}$ and inorganic carbonate concentration at SOC is equal to natural seawater at 400 ppm $CO_2$ (~2650 mg/L [$SO_4^{2-}$], ~2100 μmol/L DIC), the elevation of internal pH to 9 creates a $SO_4^{2-}$:$CO_3^{2-}$ of ~25, leading to the stabilization of vaterite and a band enriched in $SO_4^{2-}$ close to the primary organic sheet, which we observe in the chamber wall distribution of all three of our species (Fig. 5). Note that this is a maximum theoretical $SO_4^{2-}$:$CO_3^{2-}$, since DIC might be higher in the SOC, but it is very unlikely the DIC increases to a point where the ratio will be <1. TDuring precipitation, the $SO_4^{2-}$:$CO_3^{2-}$ likely decreases during the thickening of the chamber wall, due to continuous active pumping of protons out of the site of calcification (Toyofuku et al., 2017)(Toyofuku et al., 2017) (Fig. 7). This implies that the S/Ca distribution in the foraminiferal chamber wall may reflect a change in $SO_4^{2-}$/$CO_3^{2-}$ of the calcifying fluid in the site of calcification (SOC) during precipitation of the shell wall. Assuming a stable D during calcification (E.g. Dx1000= 0.013;Busenberg and Niel Plummer, 1985). $SO_4^{2-}$/$CO_3^{2-}$ at the SOC would be a factor of 3.6 higher during the thin, high-concentration band (with an S/Ca of 6.9 mmol/mol; Fig. 7) compared to the broader, low-concentration band (with an S/Ca of 1.9 mmol/mol). This decrease by a factor of 3.6 could be due to an increase in [$CO_3^{2-}$] and/or a decrease in [$SO_4^{2-}$] during precipitation. The latter

could be the result of inclusion of small amounts of sulfate in the SOC at the beginning of chamber formation and ongoing incorporation of sulfate in the foraminiferal calcite. However, since the S/Ca is not decreasing towards the outer side of the shell in the low-concentration band, the former process, i.e. increasing $CO_3^{2-}$, might be more likely. An increase of $CO_3^{2-}$ at the SOC from the first stage of chamber formation (high in S/Ca) to the broader second part (low in S/Ca) could be caused by an increase in internal pH due to proton pumping (Toyofuku et al., 2017). The band of high S/Ca would then be precipitated when proton pumping has not yet reached its maximum rate and the internal pH is still rising (Glas et al., 2013). However, to confirm this hypothesis, 
[revised manuscript text omitted]

**SUPPLEMENTARY INFORMATION**

| Species | EPMA maps total n | n specimens (n chambers) | Total n of transect maps (n of selected transect maps) |
|---|---|---|---|
| *A. tepida* | 11 | 6(12) | 24(12) |
| *B. marginata* | 8 | 4(8) | 16(8) |
| *A. lessonii* | 8 | 6(9) | 16(9) |

**Table S1: Number of maps measured by EPMA per species, with number of specimens and chambers analyzed and total number of transect maps (S/Ca and Mg/Ca values reported in Fig. 6). Note that the number of chambers analyzed is higher than the number of maps, since some maps are selected on the cross sections of two chambers (see Fig. S1-3). The last column gives the number of transects maps selected for peak-base analysis, which was limited to one per chamber to avoid overrepresenting of one chamber. The values of the peak base analysis are reported in Fig. 7 and Table 2).**

[Figure]

**Fig. S1: Overview of specimens of *Ammonia tepida* analyzed by EPMA with map selection area (blue rectangle), number of transects per map (red) and transect use for peak-base analysis (red line). Total number of maps and transect per specimen is indicated in the bottomleft corner of every SEM overview picture. Chambers formed in culture are indicated with an asterisk. Scale bar = 100µm.**

[Figure]

**Fig. S2: Overview of specimens of *Bulimina marginata* analyzed by EPMA with map selection area (blue rectangle), number of transects per map (red) and transect use for peak-base analysis (red line). Total number of maps and transect per specimen is indicated in the bottomleft corner of every SEM overview picture. Chambers formed in culture are indicated with an asterisk. Scale bar = 100µm.**

[Figure]

**Fig. S3: Overview of specimens of *Amphistegina lessonii* analyzed by EPMA with map selection area (blue rectangle), number of transects per map (red) and transect use for peak-base analysis (red line). Total number of maps and transect per specimen is indicated in the bottomleft corner of every SEM overview picture. Chambers formed in culture are indicated with an asterisk. Scale bar = 100µm.** NB, the map of top left specimen was used in Fig. 1.

[revised manuscript text omitted]

---

## Author Response (AR3)

Dear Editor,

We would like to thank both reviewers again for their second review of our manuscript. Based on the reports of reviewer 1 and 2, we finalized the manuscript. We accepted their final suggestions, including the changes recommended for Figure 9. Below you may find the final version, with the relevant changes indicated in yellow.

Sincerely,
On behalf of all co-authors,

Inge van Dijk

[revised manuscript text omitted]

For both sets of measurements, samples were measured against six ratio calibration standards with a matching matrix, i.e. 40 ppm Ca for the temperature set and 100 ppm Ca for the Burgers' Zoo set. In addition to the foraminiferal samples, we measured several standards, including NFHS-1 (NIOZ Foraminifera House Standard; for details see Mezger et al., 2016), JCt-1 (Giant Clam, *Tridacna gigas*) and JCp-1 (coral, Porites sp.; Okai et al., 2002) to monitor drift and the quality of the analyses. One of the ratio calibration standard was measured after

every 5$^{th}$ sample to monitor drift. Accuracy of Mg/Ca is 105% and 101% for JCt-1 and JCp-1, respectively with an external precision of 0.4% for both standards. Only JCp-1 has a certified value for S/Ca, and accuracy for our measurements is 94% based on this standard. The external precision of S/Ca is 1.7% and 1.0% for JCt-1 and JCp-1. We used a ratio calibration method (de Villiers et al., 2002) to calculate foraminiferal S/Ca (mmol/mol).

**2.3.4. Shell wall variability by EPMA**

Specimens of various foraminiferal species (*Ammonia tepida*, *Bulimina marginata*) from a recently published culture study (Barras et al., 2018), and *Amphistegina lessonii* cultured in the same culture set-up, were prepared for electron probe micro-analysis (EPMA) to investigate the intra-shell incorporation of sulfur and magnesium. These foraminifera were cultured under hypoxia (30% oxygen saturation) in controlled stable conditions and previously studied to investigate the Mn incorporation in foraminiferal calcite (for details and culture methodology, see Barras et al., 2018). *Ammonia tepida* and *Bulimina marginata* were cultured at 12°C, while specimens of *Amphistegina lessonii* were grown at 25°C. For the latter species, the set-up was equipped with a light system with 12 hr/12 hr light cycle."

[revised manuscript text omitted]

and S. Fig. 1). The absence of an effect of temperature on S/Ca suggests that the process responsible for increasing Mg, does not affect $SO_4^{2-}$ substitution for $CO_3^{2-}$ in the crystal lattice. Since temperature-induced changes in Mg incorporation do not increase foraminiferal S/Ca, Mg/Ca and S/Ca might therefore co-vary due to a different process, possibly by mechanisms involved in biomineralization.

**4.2. Mg and S distribution in the foraminiferal shell**

For all three species studied here we observe a positive correlation between Mg/Ca and S/Ca within chamber walls (intra-specimen variability; for an example, see Fig. 1C and last part of section 3.3.), between transects maps for each species (inter-specimen variability; Fig. 6) and between species (inter-species variability; Fig. 4). Both bands of high S (and high Mg) seem to be located close to organic linings, as shown previously for *A. gibbosa* (by EPMA; van Dijk et al., 2017a), *A. lobifera* (by electron probe WDS; Erez, 2003) and *Orbulina universa* (by nanoSIMS; Paris et al., 2014). The presence of organic material could cause a higher Mg content due to increased adsorption of Mg (Mavromatis et al., 2017). If also the case for other elements, including S, this could explain the observed covariation within chambers (Fig. 5), as earlier suggest for the planktonic foraminifer *Pulleniatina obliquiloculata* by Kunioka et al. (2006). However, this is disputed by the work of Busenberg and Plummer (1985) and Kitano et al. (1975), which shows that $SO_4^{2-}$ (as well as Na, Amiel et al., 1973) is predominately present in solid solution and not as a component of the organic matrix of biogenic (Mg-)calcites.

As shown in Fig. 1D, the relative intensity of Mg/Ca and S/Ca for different peaks in the same transect are not always similar (e.g. rightmost peaks in Mg/Ca and S/Ca are equally high, whereas the S/Ca peak in the middle of the chamber wall is much lower than that of Mg/Ca), suggesting that S and Mg are spatial (and potential temporary) correlated, but their concentrations must be partly decoupled as well. For Mg/Ca, peak and base values are higher for *A. lessonii* than for the low Mg species *A. tepida* and *B. marginata* (Fig. 7). The higher Mg/Ca in shells of *A. lessonii* compared to the two low Mg species (*A. tepida* and *B. marginata*) appears to be caused by an increase in both peak and base concentrations in *A. lessonii* specimens, already suggested by Geerken et al. (accepted). In contrast, the base values (i.e. non-band area) of S/Ca seems to be very similar for all the three species (1.5±0.1 mmol/mol for both *A. tepida* and *B. marginata* and 1.9±0.2 mmol/mol for *A. lessonii)* suggesting that the increase in S/Ca for species with higher Mg incorporation (Fig. 4) might be due to higher S-peaks in the S/Ca banding.

By comparing S/Ca and Mg/Ca values within transects we can test whether there is a correlation of S/Ca and Mg/Ca between specimens of the same species (inter-specimen variability). Individual transects were compared for each of the three species, showing a significant positive correlation between $S/Ca_{EPMA}$ and $Mg/Ca_{EPMA}$ (Fig. 6). Based on the slopes of the regression, S/Ca increases with ~37 and ~39% for *A. tepida* and *B. marginata* respectively, and ~9% for *A. lessonii* relative to Mg/Ca. The larger benthic foraminifer *A. lessonii* has the highest S incorporation, but the least sensitive S/Ca-Mg/Ca slope. In all species, higher or lower average Mg/Ca and S/Ca values are likely caused by respectively more or less intense banding between specimens (i.e. inter-species variability). The peak of bands of Mg and S of the small benthic species (*A. tepida* and *B. marginata*) are much closer, peak S/Ca are 43 and 36% of Mg/Ca respectively in these species, than for *A. lessonii* in which the S/Ca peak is 12% of the Mg/Ca peak. This results in a higher relative increase of S/Ca with Mg/Ca for the small benthic

species, and thus a steeper S/Ca-Mg/Ca slope. Still, although the slopes differ between these groups, S/Ca and Mg/Ca are consistently positively correlated.

**4.3. What controls $Mg^{2+}$ and $SO_4^{2-}$ uptake?**

Correlation between S/Ca and Mg/Ca in foraminiferal calcite (Fig. 6) might reflect i) precipitation processes, occurring at the crystal-solution interface (e.g. effects of lattice strain and crystal growth rate) or in the solution occupying the site of calcification (e.g. speciation of elements in seawater and the effect of elevated pH), ii) biomineralization-related processes, like a coupling of ion transport to the site of calcification, or iii) a combination of both. The amount of variability and unknowns in combination with the lack of knowledge on crucial processes involved in foraminiferal calcification make it challenging to assess which processes are ultimately responsible for the uptake and incorporation of both sulfate and magnesium into foraminiferal calcite. However, the observed lack of a temperature effect on S incorporation in contrast to the major impact of temperature on Mg/Ca may render some explanations more likely than others.

**4.3.1. Calcite precipitation and physico-chemical conditions**

The observed link between S/Ca and Mg/Ca might be explained by investigating parameters involved in inorganic precipitation studies. Chemical processes operating at the crystal-solution interface or in the fluid contained in the site of calcification might give insights in the observed correlation between sulfate and magnesium incorporation in foraminiferal calcite, as well as the temperature effect on Mg incorporation. Magnesium ions in the parent solution have been found to increase the co-precipitation of other elements (Okumura and Kitano, 1986). However, this is observed for alkali metal ions, which are in interstitial positions or substitute for $Ca^{2+}$ in the crystal lattice, while $SO_4^{2-}$ is hypothesized to exchange for $CO_3^{2-}$ ions (Pingitore et al., 1995; Perrin et al., 2017; Berry, 1998). Besides promoting co-precipitation, incorporation of magnesium in carbonate is suggested to cause strain on the crystal lattice, leading to distortion, and an increase in the incorporation of other elements (Mucci and Morse, 1983). This theory has been used to explain incorporation of certain elements, like $Na^+$ and $Sr^{2+}$ in larger benthic foraminifera (Evans et al., 2015). Besides the lack of study on the incorporation of sulfate with increasing Mg content, based on our data, the correlation between Mg/Ca and S/Ca cannot be explained by crystal lattice distortion. The lack of response of S/Ca to (temperature-induced) changes in Mg/Ca (Fig. 3) together with the similar base values of S/Ca for all three investigated species, while base values for Mg/Ca vary between species (Fig. 7), show that S/Ca and Mg/Ca are not always correlated, which should be the case with this hypothesis.

The effect of temperature on Mg/Ca has been comprehensively studied for inorganic carbonates, by controlled precipitation experiments (for a summary see Mucci, 1987). Over the last decades, several explanations have been proposed to explain the relation between temperature and Mg incorporation in inorganic carbonates. Firstly, the partitioning of certain elements in inorganic experiments heavily relies on precipitation rate (e.g. Lorens, 1981). This was suggested for Mg incorporation (Chilingar, 1962), which may indicate that the increase of foraminiferal Mg/Ca in our study could be explained by a positive effect of temperature on precipitation rate. However, this was disputed by several studies (e.g. Mucci and Morse, 1983; Mucci et al., 1985), showing inorganic precipitation rate does not change with temperature, but is depending on the Mg/Ca ratio of the parent solution. On the other hand, partitioning of $SO_4^{2-}$ in synthetic calcite does increase with precipitation rate (Busenberg and Plummer,

1985). However, partitioning of elements in inorganic precipitation studies might differ greatly from foraminiferal calcite, since shell precipitation rate could be decoupled from crystal precipitation rate. To discuss if S/Ca, as well as Mg/Ca of foraminiferal calcite is affected by the growth rate of foraminiferal shells, we would need to study the precipitation rate of foraminiferal calcite as a function of temperature.

It has been proposed that calcium- and magnesium transport to the site of calcification requires complete or partial dehydration of these ions, an energy-consuming process that is influenced by temperature (Mucci, 1986; Morse et al., 2007; Arvidson and Mackenzie, 2000). Re-hydration of these ions at the site of calcification (SOC) may furthermore determine isotopic fractionation during calcium carbonate precipitation (Mavromatis et al., 2013). Since dehydration of magnesium ions costs less energy at higher temperatures, it may be expected that there would be more dehydrated and transportable Mg available. This would lead to an increased (accidental) transport of $Mg^{2+}$ to the SOC by $Ca^{2+}$-pumps leading to a positive effect of temperature on Mg/Ca, or an increased selective removal of $Mg^{2+}$ resulting in theory in a lower shell Mg/Ca at higher temperatures. Since the latter is not observed, the effect of the (de)hydration of Mg ions is only likely in biomineralization models where Mg is not selectively removed, like the Trans Membrane Transport (TMT) mixing model (Nehrke et al., 2013). Although this explains the lack of a clear temperature effect on S/Ca-values, it does not explain the coregulation of S/Ca and Mg/Ca.

[revised manuscript text omitted]

**4.3.3. Sulfate at the site of calcification**

Species specific differences in the relative contribution of SWV and TMT might provide an explanation of our results. However, while this could give insights in the incorporation of S and Mg as a function of temperature and explain species-specific differences, we did not consider the inhibition effect of sulfate and the probable non-classical calcification pathway foraminifera utilize to create their shell (Jacob et al., 2017). Sulfate is a known inhibitor for precipitation of calcite (e.g. Manoli and Dalas, 2000; Kitano, 1962), but does play a role in the transformation from amorphous calcium carbonate into vaterite (see Bots et al., 2012 and references therein). Vaterite transforms into calcite via dissolution-reprecipitation when solution $SO_4^{2-}:CO_3^{2-} < 1.3$ (Fernández-Díaz et al., 2010). A recent study has proposed that certain species of planktonic foraminifera create their shell by a pathway involving vaterite phases that transform ultimately to calcite (Jacob et al., 2017), which might suggest the $SO_4^{2-}:CO_3^{2-}$ at the site of calcification is >1 when precipitation of the carbonate shell commences. Just prior to formation of a new chamber, the sulfate concentration at the SOC is probably similar to that in seawater, assuming the calcification fluid is composed of either a small volume of seawater enclosed by the protective envelop separating the SOC from seawater or by seawater vacuoles (SWV model). With a seawater concentration of ~30 mM $[SO_4^{2-}]$, it is very likely that the $SO_4^{2-}:CO_3^{2-}$ at the site of calcification is > 1, but it is depending of the carbonate chemistry at the SOC. Laboratory experiments have revealed that the internal pH of a foraminifera is elevated, (species-specific) values ranging from ~8.75 (Bentov et al., 2009) to ≥9 (de Nooijer et al., 2009) at the start of shell formation due to proton pumping (Toyofuku et al., 2017), which lowers the pH in the microenvironment surrounding the foraminifer (Glas et al., 2012). When assuming the $SO_4^{2-}$ and inorganic carbonate concentration at SOC is equal to natural seawater at 400 ppm $CO_2$ (~2650 mg/L $[SO_4^{2-}]$, ~2100 µmol/L DIC), the elevation of internal pH to 9 creates a $SO_4^{2-}:CO_3^{2-}$ of ~25, leading to the stabilization of vaterite and a band enriched in $SO_4^{2-}$ close to the primary organic sheet, which we observe in the chamber wall distribution of all three of our species (Fig. 5). Note that this is a maximum theoretical $SO_4^{2-}:CO_3^{2-}$, since DIC might be higher

in the SOC, but it is very unlikely that the DIC increases to a point where the ratio will be <1. During precipitation, the $SO_4^{2-}$:$CO_3^{2-}$ likely decreases during the thickening of the chamber wall, due to continuous active pumping of protons out of the site of calcification (Toyofuku et al., 2017). This implies that the S/Ca distribution in the foraminiferal chamber wall may reflect a change in $SO_4^{2-}$/$CO_3^{2-}$ of the calcifying fluid in the site of calcification (SOC) during precipitation of the shell wall. Assuming a stable D during calcification (E.g. Dx1000= 0.013; Busenberg and Plummer, 1985), $SO_4^{2-}$/$CO_3^{2-}$ at the SOC would be a factor of 3.6 higher during the thin, high-concentration band (with an S/Ca of 6.9 mmol/mol; Fig. 7) compared to the broader, low-concentration band (with an S/Ca of 1.9 mmol/mol). This decrease by a factor of 3.6 could be due to an increase in $[CO_3^{2-}]$ and/or a decrease in $[SO_4^{2-}]$ during precipitation. The latter could be the result of inclusion of small amounts of sulfate in the SOC at the beginning of chamber formation and ongoing incorporation of sulfate in the foraminiferal calcite. However, since the S/Ca is not decreasing towards the outer side of the shell in the low-concentration band, the former process, i.e. increasing $CO_3^{2-}$ , might be more likely. An increase of $CO_3^{2-}$ at the SOC from the first stage of chamber formation (high in S/Ca) to the broader second part (low in S/Ca) could be caused by an increase in internal pH due to proton pumping (Toyofuku et al., 2017). The band of high S/Ca would then be precipitated when proton pumping has not yet reached its maximum rate and the internal pH is still rising (Glas et al., 2013). However, to confirm this hypothesis, a more precise characterization of the calcification fluid's chemistry is necessary.

**5. Conclusions**

[revised manuscript text omitted]